# Optimal Gradient Sliding and its Application to Distributed Optimization Under Similarity

**Dmitry Kovalev**
KAUST,* Saudi Arabia
dakovalev1@gmail.com

**Aleksandr Beznosikov**
MIPT,† HSE University and Yandex, Russia
anbeznosikov@gmail.com

**Ekaterina Borodich**
MIPT and HSE University, Russia
borodich.ed@phystech.edu

**Alexander Gasnikov**
MIPT, HSE University and IITP RAS,‡ Russia
gasnikov@yandex.ru

**Gesualdo Scutari**
Purdue University, USA
gscutari@purdue.edu

## Abstract

We study structured convex optimization problems, with additive objective $r :=
p + q$, where $r$ is ($\mu$-strongly) convex, $q$ is $L_q$-smooth and convex, and $p$ is $L_p$-smooth, possibly nonconvex. For such a class of problems, we proposed an inexact accelerated gradient sliding method that can skip the gradient computation for one of these components while still achieving optimal complexity of gradient calls of $p$ and $q$, that is, $\mathcal{O}(\sqrt{L_p/\mu})$ and $\mathcal{O}(\sqrt{L_q/\mu})$, respectively. This result is much sharper than the classic black-box complexity $\mathcal{O}(\sqrt{(L_p + L_q)/\mu})$, especially when the difference between $L_p$ and $L_q$ is large. We then apply the proposed method to solve distributed optimization problems over master-worker architectures, under agents' function similarity, due to statistical data similarity or otherwise. The distributed algorithm achieves for the first time lower complexity bounds on *both* communication and local gradient calls, with the former having being a long-standing open problem. Finally the method is extended to distributed saddle-problems (under function similarity) by means of solving a class of variational inequalities, achieving lower communication and computation complexity bounds.

## 1 Introduction

We consider structured convex programming in the form [6, 13, 35]:

$$\min_{x\in\mathbb{R}^d} r(x) := q(x) + p(x), \tag{1}$$

where $r$ is assumed to be convex and decomposed as the sum of a smooth, possibly nonconvex function $p$ and a smooth convex function $q$. First order information of $p$ and $q$ is accessible separately. We are interested in scenarios where the cost of evaluating the gradient of the two functions is not even, but computing, say $\nabla p$, is much more resource demanding than $\nabla q$. The motivating application for this scenario is distributed optimization over master-worker systems, as discussed next.

---

*King Abdullah University of Science and Technology

†Moscow Institute of Physics and Technology

‡Institute for Information Transmission Problems RAS

36th Conference on Neural Information Processing Systems (NeurIPS 2022).

Consider the following distributed optimization problem over a network of $n$ agents:

$$\min_{x \in \mathbb{R}^d} r(x) = \frac{1}{n} \sum_{i=1}^{n} f_i(x), \tag{2}$$

where $f_i$ is the loss function of agent $i$, assumed to be convex, which is not known to other agents. Agents are embedded in a star-topology, with agent 1 being the master node, without loss of generality–this is the typical federated learning setup [24]. An instance of (2) of particular interest is the empirical risk minimization (ERM) whereby the goal is to minimize the average loss over some dataset, distributed across the nodes of the network, with $f_i$ being the empirical risk of agent $i$, i.e., $f_i(x) = \frac{1}{m} \sum_{j=1}^{m} \ell(x; z_i^j)$, where $z_i^{(1)}, \ldots, z_i^{(m)}$ is the set of $m$ samples owned by agent $i$, and $\ell(x; z_i^j)$ measures the loss of the model $x$ on the sample $z_i^j$.

Several solutions methods have been proposed to solve (2); the prototype approach consists in interleaving local computations at the workers sides (nodes $i = 1, \ldots n$) with communications to/from the master node ($i = 1$), which maintains and updates the authoritative copy of the optimization variables, producing eventually the final solution estimate. Since the cost of communications is often the bottleneck in distributed computing (e.g., [7, 32]), a lot of research has been devoted to designing distributed algorithms that are *communication efficient*. Acceleration (in the sense of Nesterov) has been extensively investigated as a procedure to reduce the communication burden. For $L$-smooth and $\mu$-strongly convex functions $r$ in (2), linear convergence is certified by employing first-order methods, with computation (gradient evaluations) and communication complexities proportional to $\sqrt{\kappa}$ ($\kappa \triangleq L/\mu$ is the condition number of $r$). For ill-conditioned functions ($\kappa$ very large), the polynomial dependence on $\kappa$ may be unsatisfactory. This is, e.g., the typical setting of many ERM problems wherein the optimal regularization parameter for test predictive performance is very small.

Further improvements on the communication complexity can be obtained exploiting the extra structure typical in ERM problems, also known as *function similarity* (see, e.g., [4, 51, 44, 48]): $\|\nabla^2 f_i(x) - \nabla^2 f_j\| \leq \delta$, for all $x$ in a proper domain of interest and all $i \neq j = 1, \ldots, n$, where $\delta > 0$ measures the degree of similarity between the Hessian matrices of the local losses. When data are i.i.d. among agents, $f_i$'s reflect statistical similarities in local data, resulting in $\delta = \tilde{O}(1/\sqrt{m})$ with high-probability ($\tilde{O}$ hides log-factors and dependence on $d$). In this scenario, in general, $1 + \delta/\mu \ll \kappa$ [4]. This motivated a surge of studies aiming at exploiting function similarity coupled with acceleration to boost communication efficiency (see Sec. 1.2 for an overview of relevant works): linear convergence is certified with a number of communication steps (for nonquadratic losses) scaling with $\widetilde{\mathcal{O}}(\sqrt{\delta/\mu})$, where $\widetilde{\mathcal{O}}$ hides log-factors. This matches lower (communication) complexity bounds [4] ıt only up to log-factors. Furthermore, these methods are not computationally optimal, yielding local gradients calls larger than lower complexity bounds $\mathcal{O}(\sqrt{\kappa})$. In fact, Table 1 shows that, to the date, there exists no distributed algorithm achieving the best of the two worlds, that is, optimal (lower bound) communication complexity *and* local gradient (oracle) complexity.

This paper fills this gap. Our starting point is the reformulation of (2) in the equivalent form

$$\min_{x \in \mathbb{R}^d} r(x) = \underbrace{f_1(x)}_{:=q(x)} + \underbrace{\frac{1}{n} \sum_{i=1}^{n} [f_i(x) - f_1(x)]}_{:=p(x)}, \tag{3}$$

which exploits function similarity at the agents' side via preconditioning. Problem (3) is an instance of (1): all $f_i$ (thus $q$) are convex but $p$ is nonconvex. Also, evaluating $\nabla q$ and $\nabla p$ has different costs; the former involves only local computations at the master node while the latter requires communications from/to master and workers nodes. At high-level the idea is then clear: one would like to design a distributed algorithm for (3) [or more generally for (1)] that skips gradient computations of $\nabla p$ (saving thus communications) without slowing down the overall optimal rate of convergence.

This naturally suggests the use of *gradient-sliding* techniques [19, 41, 10], yielding algorithms that skip from time to time computation of the gradient of one function in the summand objective. However, existing gradient-sliding algorithms are not applicable to (1) [and thus (3)] because they all require $p$ and $q$ to be *convex*. This calls for new designs, accounting for the nonconvexity of $p$.

## 1.1 Main contributions

Our contribution is threefold:

• **A new gradient-sliding algorithm for** (1)**:** We propose a new Accelerated ExtraGradient sliding method that skips the computation of $\nabla p$ from time to time. The method builds on an inexact acceleration of a proximal envelop (outer-loop) coupled with a suitable termination criterion and inner-loop algorithm to approximately solve the proximal subproblem. When applied to (1), with $r$ being ($\mu$-strongly) convex, $q$ being $L_q$-smooth and convex, and $p$ being $L_p$-smooth (possibly nonconvex), the proposed algorithm achieves optimal complexity of gradient calls of $q$ and $p$, that is,

| $r$ | $\nabla q$ | $\nabla p$ |
|---|---|---|
| strongly convex | $\mathcal{O}\left(\sqrt{\frac{L_q}{\mu}}\log\frac{1}{\varepsilon}\right)$ | $\mathcal{O}\left(\sqrt{\frac{L_p}{\mu}}\log\frac{1}{\varepsilon}\right)$ |
| convex | $\mathcal{O}\left(\sqrt{\frac{L_q}{\varepsilon}}\|x^0-x^*\|\right)$ | $\mathcal{O}\left(\sqrt{\frac{L_p}{\varepsilon}}\|x^0-x^*\|\right)$ |

Notice that the above complexity bounds are sharper than the complexity bound obtained by the Nesterov's optimal first-order method for smooth (strongly) convex optimization applied to (1). For instance for strongly convex $r$, that would yield $\nabla q, \nabla p$ complexity scaling as $\mathcal{O}(\sqrt{(L_q + L_p)/\mu})$, which is less favorable than our separate complexity bounds above. To the best of our knowledge, this is the first time that such bounds are achieved for nonconvex $p$.

• **Optimal complexity bounds for** (2) **under function similarity:** We customize the proposed accelerated gradient-sliding algorithm to the distributed optimization problem (2) under $\delta$-function similarity. As showed in Table 1 for strongly convex $r$ (see Sec. 3 for the case of convex $r$), the new distributed algorithm achieves lower complexity bounds on *both* the number of communications [4] and on the number of gradient computations (without logarithmic factors!) [37]. Achieving optimal communication complexity (for nonquadratic losses) was a long-standing open problem.

• **Gradient sliding for variational inequalities:** We extend the proposed gradient-sliding machinery to solve distributed saddle-points under similarity by means of solving a class of strongly-monotone Variational Inequalities (VI). We improve existing complexity bounds for such problems [9] achieving for the first time both optimal communication complexity and gradient oracle complexity–see Table 1.

Table 1: Existing convergence results for distributed (saddle point) optimization under $\delta$-similarity.

| | | Reference | Communication complexity | Local gradient complexity | Order | Limitations |
|---|---|---|---|---|---|---|
| Minimization | Upper | DANE [42] | $\mathcal{O}\left(\frac{\delta^2}{\mu^2}\log\frac{1}{\varepsilon}\right)$ | $\mathcal{O}\left(\sqrt{\frac{L}{\mu}}\sqrt{\frac{\delta^3}{\mu^3}}\log^2\frac{1}{\varepsilon}\right)$ [2] | 1st | quadratic |
| | | DiSCO [51] | $\mathcal{O}\left(\sqrt{\frac{\delta}{\mu}}(\log\frac{1}{\varepsilon}+C^2\Delta F_0)\log\frac{L}{\mu}\right)$ | $\mathcal{O}\left(\sqrt{\frac{L}{\mu}}(\log\frac{1}{\varepsilon}+C^2\Delta F_0)\log\frac{L}{\mu}\right)$ | 2nd | $C$ - self-concordant [3] |
| | | AIDE [40] | $\mathcal{O}\left(\sqrt{\frac{\delta}{\mu}}\log\frac{1}{\varepsilon}\log\frac{L}{\delta}\right)$ | $\mathcal{O}\left(\sqrt{\frac{L}{\mu}}\sqrt{\frac{\delta}{\mu}}\log\frac{1}{\varepsilon}\log\frac{L}{\delta}\right)$ [4] | 1st | quadratic |
| | | DANE-LS [50] | $\mathcal{O}\left(\frac{\delta}{\mu}\log\frac{1}{\varepsilon}\right)$ | $\mathcal{O}\left(\sqrt{\frac{L}{\mu}}\frac{\delta^{3/2}}{\mu^{3/2}}\log\frac{1}{\varepsilon}\right)$ [5] | 1st/2nd | quadratic [6] |
| | | DANE-HB [50] | $\mathcal{O}\left(\sqrt{\frac{\delta}{\mu}}\log\frac{1}{\varepsilon}\right)$ | $\mathcal{O}\left(\sqrt{\frac{L}{\mu}}\frac{\delta}{\mu}\log\frac{1}{\varepsilon}\right)$ [5] | 1st/2nd | quadratic [6] |
| | | SONATA [45] | $\mathcal{O}\left(\frac{\delta}{\mu}\log\frac{1}{\varepsilon}\right)$ | $\mathcal{O}\left(\sqrt{\frac{L}{\mu}}\sqrt{\frac{\delta}{\mu}}\log^2\frac{1}{\varepsilon}\right)$ [2] | 1st | decentralized |
| | | SPAG [21] | $\mathcal{O}\left(\sqrt{\frac{L}{\mu}}\log\frac{1}{\varepsilon}\right)$ [1] | $\mathcal{O}\left(\sqrt{\frac{L}{\mu}}\sqrt{\frac{L}{\delta}}\log^2\frac{1}{\varepsilon}\right)$ [1,2] | 1st | $M$ - Lipshitz hessian |
| | | DiRegINA [12] | $\mathcal{O}\left(\frac{\delta}{\mu}\log\frac{1}{\varepsilon}+\sqrt{\frac{M\delta R_0}{\mu}}\right)$ | $\mathcal{O}\left(\sqrt{\frac{L}{\mu}}\sqrt{\frac{\delta}{\mu}}\log^2\frac{1}{\varepsilon}+\sqrt{\frac{MLR_0}{\mu}}\log\frac{1}{\varepsilon}\right)$ [2] | 2nd | $M$ -Lipshitz hessian |
| | | ACN [1] | $\mathcal{O}\left(\sqrt{\frac{\delta}{\mu}}\log\frac{1}{\varepsilon}+\sqrt[3]{\frac{M\delta R_0}{\mu}}\right)$ | $\mathcal{O}\left(\sqrt{\frac{L}{\mu}}\log^2\frac{1}{\varepsilon}+\sqrt[3]{\frac{M\delta R_0}{\mu}}\sqrt{\frac{L}{\delta}}\log\frac{1}{\varepsilon}\right)$ [2] | 2nd | $M$ -Lipshitz hessian |
| | | AccSONATA [46] | $\mathcal{O}\left(\sqrt{\frac{\delta}{\mu}}\log\frac{1}{\varepsilon}\log\frac{L}{\mu}\right)$ | $\mathcal{O}\left(\sqrt{\frac{L}{\mu}}\log^2\frac{1}{\varepsilon}\log\frac{\delta}{\mu}\right)$ [2] | 1st | decentralized |
| | | This paper | $\mathcal{O}\left(\sqrt{\frac{\delta}{\mu}}\log\frac{1}{\varepsilon}\right)$ | $\mathcal{O}\left(\sqrt{\frac{L}{\mu}}\log\frac{1}{\varepsilon}\right)$ | 1st | |
| | Lower | [4] | $\mathcal{O}\left(\sqrt{\frac{\delta}{\mu}}\log\frac{1}{\varepsilon}\right)$ | — | | |
| | | [37] | — | $\mathcal{O}\left(\sqrt{\frac{L}{\mu}}\log\frac{1}{\varepsilon}\right)$ | | non-distributed |
| Saddles | Upper | SMMDSA [9] | $\mathcal{O}\left(\frac{\delta}{\mu}\log\frac{1}{\varepsilon}\right)$ | $\mathcal{O}\left(\frac{L}{\mu}\log\frac{1}{\varepsilon}\log\frac{L}{\mu}\right)$ | 1st | |
| | | This paper | $\mathcal{O}\left(\frac{\delta}{\mu}\log\frac{1}{\varepsilon}\right)$ | $\mathcal{O}\left(\frac{L}{\mu}\log\frac{1}{\varepsilon}\right)$ | 1st | |
| | Lower | [9] | $\mathcal{O}\left(\frac{\delta}{\mu}\log\frac{1}{\varepsilon}\right)$ | — | | |
| | | [39] | - | $\mathcal{O}\left(\frac{L}{\mu}\log\frac{1}{\varepsilon}\right)$ | | non-distributed |

[1] This is the worst-case complexity, as pointed out in the paper; the convergence of the method might be better, based upon an additional sequence $G_t$ [21]; [2] proximal local computations (exact solution of local subproblems), we filled these cells assuming that the proximal operator is computed using Accelerated Gradient Descent [37] with accuracy $\varepsilon^2$; [3] from Lipschitzness of the Hessian and strong convexity follows self-concordance; [4] gradient complexity not provided, we derived it using [37]; [5] gradient complexity not provided, we derived it using [38]; [6] gradient complexity holds for nonquadratic functions;
*Notation:* $\delta$ = similarity parameter, $L$=smoothness constant of $f_i$, $\mu$ = strong convexity constant of $r$, $\varepsilon$ =accuracy of the solution, $R_0 := \|x^0 - x^*\|$, $\Delta F_0 := r(x^0) - r(x^*)$.

## 1.2 Related works

**Gradient-Sliding:** Since the seminal paper [28], the idea of gradient-sliding for structured convex optimization such as (1) has received significant attention as tool to skip gradient computations of one function in the summand; examples and generalization include first-order accelerated methods [28, 29, 31], zero-order (derivative-free) schemes [14, 22, 8, 43], high-order methods [25, 17, 2, 20], and slidings for saddle point problems and variational inequalities [3, 30, 47, 9]. Albeit applicable to more general classes of optimization problems than (1) (e.g., allowing either $p$ and $q$ to be nonsmooth), none of the existing methods provide guarantees when $p$ is nonconvex. The proposed algorithm fills this gap. Furthermore, it achieves optimal lower complexity bounds on the calls of $\nabla p$ and $\nabla q$.

**Distributed optimization under function similarity:** The literature of distributed optimization is vast; given the focus of this work, we comment next solution methods exploiting function similarity via proper preconditioning–Table 1 summarizes complexity results of existing distributed methods solving either minimization problems or saddle-point formulations, and is commented next.

The seminal paper [4] established lower communication complexity bounds for (2) under $\delta$-similarity: $\varepsilon$-optimality cannot be achieve in less than $\Omega(\sqrt{\delta/\mu}\log 1/\varepsilon)$ communication rounds. Since then, a lot of effort has been devoted to design distributed schemes aiming at achieving optimal communication complexity. The authors in [42] proposed DANE, a mirror-descent based algorithm whereby workers perform a local data preconditioning via a suitably chosen Bregman divergence, and the master averages the solutions of the workers. For *quadratic* losses, DANE achieves communication complexity $\widetilde{\mathcal{O}}((\delta/\mu)^2\log 1/\varepsilon)$; this was later improved to $\mathcal{O}((\delta/\mu)\log 1/\varepsilon)$ for nonquadratic losses in [45], where the SONATA algorithm was proposed (also implementable over mesh-networks).

Improvements were achieved employing acceleration; efforts include: DiSCO [51], an inexact damped Newton method coupled with a preconditioned conjugate gradient (to compute the Newton direction), which achieves communication complexity $\widetilde{\mathcal{O}}(\sqrt{\delta/\mu})\log 1/\varepsilon$ for self-concordant losses (see Table 1 for the log-factors hidden in the $\widetilde{\mathcal{O}}$); AIDE [40], which uses the Catalyst framework [33], matching the rate of DiSCO for quadratic losses; DANE-HB [50], a variant of DANE equipped with Heavy Ball momentum and matching for quadratic functions the communication complexity of DiSCO and AIDE; and SPAG [21], a preconditioned direct accelerated method, achieving for nonquadradic losses *asymptotically* the convergence rate $\mathcal{O}((1-1/\sqrt{\beta/\mu})^k)$ ($k$ is the iteration index)–the worst-case rate is still $\mathcal{O}(\sqrt{L/\mu})\log 1/\varepsilon$.

Finally, higher order methods employing preconditioning have been studied in [12, 1, 46]: [12] proposed DiRegINA, a decentralization of the cubic regularization of the Newton method, where workers build Newton direction sampling local Hessians; [1] introduced ACN, an inexact accelerated cubic-regularized Newton's method, with improved complexity with respect to [12]; and [46] extended the Catalyst framework [33] to the distributed setting (including mesh networks), proposing Acc SONATA–the communication complexity of these methods is reported in Table 1.

In summary, the above tour on the relevant literature shows that none of the existing methods can match lower communication complexity bounds for (non quadratic) optimization problems (2) under function similarity (all complexity bounds contain log-factors). The proposed distributed method achieves lower communication and computation complexity bounds.

## 2 Optimal Gradient Sliding for Minimization Problems

We study the minimization problem (1), under the following blanket assumptions.

**Assumption 1.** $r(x)\colon \mathbb{R}^d \to \mathbb{R}$ *is $\mu$-strongly convex on $\mathbb{R}^d$.*

**Assumption 2.** $q(x)\colon \mathbb{R}^d \to \mathbb{R}$ *is convex and $L_q$-smooth on $\mathbb{R}^d$.*

**Assumption 3.** $p(x)\colon \mathbb{R}^d \to \mathbb{R}$ *is $L_p$-smooth on $\mathbb{R}^d$.*

The proposed Accelerated ExtraGradient sliding is formally introduced in Algorithm 1. Convergence of the outer loop is established in Theorem 1, while Theorem 2 determines complexity of solving the inner loop up to a suitable termination. Finally Theorem 3 provides the overall complexity merging inner and outer loop results. The proof of all the theorems can be found in Appendix A.1.

---

**Algorithm 1** Accelerated Extragradient

---

1: **Input:** $x^0 = x_f^0 \in \mathbb{R}^d$
2: **Parameters:** $\tau \in (0, 1], \eta, \theta, \alpha > 0, K \in \{1, 2, \ldots\}$
3: **for** $k = 0, 1, 2, \ldots, K - 1$ **do**
4: $\quad x_g^k = \tau x^k + (1 - \tau)x_f^k$
5: $\quad x_f^{k+1} \approx \arg\min_{x \in \mathbb{R}^d} \left[ A_\theta^k(x) := p(x_g^k) + \langle \nabla p(x_g^k), x - x_g^k \rangle + \frac{1}{2\theta}\|x - x_g^k\|^2 + q(x) \right]$
6: $\quad x^{k+1} = x^k + \eta\alpha(x_f^{k+1} - x^k) - \eta\nabla r(x_f^{k+1})$
7: **end for**
8: **Output:** $x^K$

---

**Theorem 1.** *Consider Algorithm 1 for Problem 1 under Assumptions 1-3, with the following tuning:*

$$\tau = \min\left\{1, \frac{\sqrt{\mu}}{2\sqrt{L_p}}\right\}, \quad \theta = \frac{1}{2L_p}, \quad \eta = \min\left\{\frac{1}{2\mu}, \frac{1}{2\sqrt{\mu L_p}}\right\}, \quad \alpha = \mu;$$

*and let $x_f^{k+1}$ in line 5 satisfy*

$$\|\nabla A_\theta^k(x_f^{k+1})\|^2 \leq \frac{L_p^2}{3}\|x_g^k - \arg\min_{x \in \mathbb{R}^d} A_\theta^k(x)\|^2. \tag{4}$$

*Then, for any*

$$K \geq 2\max\left\{1, \sqrt{\frac{L_p}{\mu}}\right\} \log \frac{\|x^0 - x^*\|^2 + \frac{2\eta}{\tau}\left[r(x^0) - r(x^*)\right]}{\varepsilon}, \tag{5}$$

*we have the following estimate for the distance to the solution $x^*$:*

$$\|x^K - x^*\|^2 \leq \varepsilon. \tag{6}$$

### 2.1 Solving the auxiliary subproblems

At each iteration of Algorithm 1, one needs to solve the subproblem:

$$\min_{x \in \mathbb{R}^d} A_\theta^k(x) := p(x_g^k) + \langle \nabla p(x_g^k), x - x_g^k \rangle + \frac{1}{2\theta}\|x - x_g^k\|^2 + q(x). \tag{7}$$

According to Theorem 1, (7) need not be solved to arbitrary precision; inexact solutions $x_f^{k+1}$ satisfying condition (4) suffice. Condition (4) means that gradient norm $\|\nabla A_\theta^k(x_f^{k+1})\|$ should be sufficiently small. Notice that $A_\theta^k(x)$ in (7) is $(2L_p + L_q)$-smooth and convex.

Problem 7 can be solved up to the termination (4) using any of the algorithms in [27, 26, 38]. We obtain the following complexity.

**Theorem 2** ([38] Remark 1). *There exists a certain algorithm such that, when applied to problem (7) with the starting point $x_g^k$, returns $x_f^{k+1}$ satisfying*

$$\|\nabla A_\theta^k(x_f^{k+1})\| \leq \frac{D^2 \cdot \max\{L_p, L_q\}\|x_g^k - \arg\min_{x \in \mathbb{R}^d} A_\theta^k(x)\|}{T^2}, \tag{8}$$

*where $D > 0$ is some universal constant (independent of $L_p, L_q, T$ etc) and $T$ is the number of calls of $\nabla q$ by the algorithm.*

### 2.2 Overall complexity of the optimal gradient-sliding

Theorem 2 suggests that, to satisfy condition (4) in Theorem 1, it is sufficient to choose the number $T$ of iterations of the inner algorithm as

$$T = \sqrt[4]{3}D\max\left\{1, \sqrt{\frac{L_q}{L_p}}\right\}. \tag{9}$$

We can now determine the overal complexity of Algorithm 1. At each iteration of Algorithm 1 we call $\nabla p$ twice (at $x_g^k$ – line 5 and at $x_f^{k+1}$ – line 6), and $\nabla q$ is computed $T + 1$ times ($T$ times in the

auxiliary problem – line 5, and at $x_f^{k+1}$ – line 6). Hence, to find an $\varepsilon$-solution of problem (1), i.e., to find $x^K \in \mathbb{R}^d$ that satisfies (6), Algorithm 1 requires $K$ iterations as given in (5),

$$2 \times K = \mathcal{O}\left(\max\left\{1, \sqrt{\tfrac{L_p}{\mu}}\right\} \log \tfrac{1}{\varepsilon}\right) \quad \text{calls of } \nabla p(x), \text{ and}$$

$$(T+1) \times K = \mathcal{O}\left(\max\left\{1, \sqrt{\tfrac{L_q}{L_p}}, \sqrt{\tfrac{L_p}{\mu}}, \sqrt{\tfrac{L_q}{\mu}}\right\} \log \tfrac{1}{\varepsilon}\right) \quad \text{calls of } \nabla q(x).$$

Putting everything together we obtain the following final convergence result.

**Theorem 3.** *Consider Problem* (1) *under Assumptions 1 to 3, with $\mu \le L_p \le L_q$. Then, to reach an $\varepsilon$-solution, Algorithm 1 requires*

$$\mathcal{O}\left(\sqrt{\tfrac{L_q}{\mu}} \log \tfrac{1}{\varepsilon}\right) \quad \text{calls of } \nabla q(x) \quad \text{and} \quad \mathcal{O}\left(\sqrt{\tfrac{L_p}{\mu}} \log \tfrac{1}{\varepsilon}\right) \quad \text{calls of } \nabla p(x).$$

This matches optimal complexity for the individual gradient calls.

We conclude this section providing convergence of variant of the proposed algorithm suitable for convex $r$ in (1) (Assumption 1 with $\mu = 0$). The algorithm is described in Appendix A.2. Here we only provide the final convergence result, the analogous of Theorem 3.

**Theorem 4.** *Consider Problem* (1) *under Assumptions 1 (with $\mu = 0$)-3, with $L_p \le L_q$. Then, to find an $\varepsilon$-solution of* (1) *(in objective value), Algorithm 3 in Appendix A.2 requires*

$$\mathcal{O}\left(\sqrt{\tfrac{L_q}{\varepsilon}} \|x^0 - x^*\|\right) \quad \text{calls of } \nabla q(x) \quad \text{and} \quad \mathcal{O}\left(\sqrt{\tfrac{L_p}{\varepsilon}} \|x^0 - x^*\|\right) \quad \text{calls of } \nabla p(x).$$

## 3 Application to Distributed Optimization Under Similarity

In this section, we apply the proposed algorithm the the distributed optimization problem (2), under the following assumptions.

**Assumption 4.** *Each $f_i(x) \colon \mathbb{R}^d \to \mathbb{R}$ is convex and $L$-smooth.*

**Assumption 5.** *$r(x) \colon \mathbb{R}^d \to \mathbb{R}$ is $\mu$-strongly convex.*

**Assumption 6.** *$f_1(x), \ldots, f_n(x)$ are $\delta$-related: $\|\nabla^2 f_i(x) - \nabla^2 f_j(x)\| \le \delta$, for all $i \ne j$ and $x \in \mathbb{R}^d$, and some $\delta > 0$.*

From the last assumption it is easy to get that for all $i$ and $x \in \mathbb{R}^d$ we have $\|\nabla^2 f_i(x) - \nabla^2 r(x)\| = \|\frac{1}{n}\sum_{j=1}^n [\nabla^2 f_i(x) - \nabla^2 f_j(x)]\| \le \frac{1}{n}\sum_{j=1}^n \|\nabla^2 f_i(x) - \nabla^2 f_j(x)\| \le \delta$.

We leverage now Algorithm 1 to solve (2), using the equivalent reformulation (3). The algorithm applied to the distributed system can be described as follows. The server computes $x_g^k$ and sends it to all the workers (line 4). Workers compute $\nabla f_i(x_g^k)$ and send it to the server. After collecting all $\nabla f_i(x_g^k)$, the server builds $\nabla p(x_g^k) = \nabla r(x_g^k) - \nabla f_1(x_g^k)$, and then solves (inexactly) the local problem $A_\theta^k$ (line 5). The inexact solution $x_f^{k+1}$ is then broadcast to the workers, which update their own receives $\nabla f_i(x_f^{k+1})$ and send back to the server, which can then evaluate $\nabla r(x_f^{k+1})$ (line 6).

Using Assumptions 4 and 5, we infer that $r$ is $\mu$-strongly convex; and $q = f_1$ is $L_q$-smooth and convex, with $L_q = L$. It follows from Assumption 6 that $\|\nabla^2 p\| \le \delta$. Therefore, $p$ has $L_p$-Lipschitz gradient, with $L_p = \delta$. This shows that we can leverage Theorems 3 and 4 to establish convergence for strongly convex and convex $r$, as given next.

**Theorem 5.** *Let Assumptions 4 to 6 be satisfied with $\mu \le \delta \le L$. Then, to find $\varepsilon$-solution of the distributed optimization problem* (2) *Algorithm 1 requires*

$$\mathcal{O}\left(\sqrt{\tfrac{\delta}{\mu}} \log \tfrac{1}{\varepsilon}\right) \quad \text{communication rounds and } \mathcal{O}\left(\sqrt{\tfrac{L}{\mu}} \log \tfrac{1}{\varepsilon}\right) \quad \text{local gradient computations.}$$

**Theorem 6.** *Let Assumptions 4, 5 (with $\mu = 0$), 6 be satisfied and $\delta \le L$. Then, to find $\varepsilon$-solution of the distributed optimization problem* (2) *Algorithm 1 requires*

$$\mathcal{O}\left(\sqrt{\tfrac{\delta}{\varepsilon}} \|x^0 - x^*\|\right) \quad \text{communication rounds and } \mathcal{O}\left(\sqrt{\tfrac{L}{\varepsilon}} \|x^0 - x^*\|\right) \quad \text{local computations.}$$

Such estimates are optimal from both communications [4] and local computations point of views [37]. It is important to remark that Algorithm 1 solves the local subproblem $A_\theta$ with some precision, while most of existing works (see Table 1) assume that local problems are solved with infinite precision (column Local gradient complexity), which is not practical. Note also that the subproblems in line 5 of Algorithm 1 do not necessarily have to be solved by a deterministic algorithm as in Theorem 2. Stochastic methods can also be used, as long as they guarantee that condition (4) is met.

# 4 Optimal Gradient Sliding for VIs

In this section we consider the composite variational inequality [16, 5] in the form:

$$\text{Find } x^* \in \mathbb{R}^d \ : \ R(x^*) = 0 \text{ with } R(x) := Q(x) + P(x), \tag{10}$$

where $Q(x), P(x) \colon \mathbb{R}^d \to \mathbb{R}^d$. Variational inequalities are a unified umbrella for a variety of problems–two examples follow.

**Example 1 [Minimization].** Consider problem (1), choose $Q(x) = \nabla q(x)$ and $P(x) = \nabla p(x)$. Then the solution of the variational inequality (10) means that we need to find the point $x^*$ where the operator $R(x^*) = \nabla r(x^*)$. For the convex function $r$, this is equivalent to finding the minimum.

**Example 2 [Saddle point problems].** Consider the convex-concave saddle point problem

$$\min_{y \in \mathbb{R}^{d_y}} \max_{z \in \mathbb{R}^{d_z}} r(y,z) := q(y,z) + p(y,z). \tag{11}$$

If we take $Q(x) := Q(y,z) = [\nabla_y q(y,z), -\nabla_z q(y,z)]$ and $P(x) := P(y,z) = [\nabla_y p(y,z), -\nabla_z p(y,z)]$, then it can be proved that $x^* = (y^*, z^*)$ is a solution for (10) if and only if $x^* = (y^*, z^*)$ is a solution for (11), i.e.

$$r(y^*, z) \le r(y^*, z^*) \le r(y, z^*) \quad \text{for all } y \in \mathbb{R}^{d_y} \text{ and } z \in \mathbb{R}^{d_z}.$$

While minimization problems are widely considered separately from variational inequalities, saddle point problems are often analyzed under the VI lens. In recent years the popularity of saddles has grown, this is due to the fact that they have both classical [15] and new ML [18, 34] applications.

We study problem (10) under the following assumptions.

**Assumption 7.** $R(x)$ is $\mu$-strongly monotone: $\langle R(x_1) - R(x_2), x_1 - x_2 \rangle \ge \mu \|x_1 - x_2\|^2$, for all $x_1, x_2 \in \mathbb{R}^d$.

**Assumption 8.** $Q(x)$ is monotone and $L_q$-Lipschitz: $\langle Q(x_1) - Q(x_2), x_1 - x_2 \rangle \ge 0$ and $\|Q(x_1) - Q(x_2)\| \le L_q \|x_1 - x_2\|$, for all $x_1, x_2 \in \mathbb{R}^d$.

**Assumption 9.** $P(x)$ is $L_p$-Lipschitz: $\|P(x_1) - P(x_2)\| \le L_p \|x_1 - x_2\|$ for all $x_1, x_2 \in \mathbb{R}^d$.

For saddle point problems these assumptions are equivalent to (strong) convexity–(strong) concavity and Lipschitzness of gradients.

## 4.1 Sliding via Extragradient

This algorithm is a non-accelerated version of Algorithm 1. A similar non-accelerated sliding is used in [9]. Our version however has better theoretical and practical guarantees because of the effective stopping criterion (12). Convergence of the outer loop is established in Theorem 7; Theorem 8 establishes convergence of the inner loop up to the required termination; and finally Theorem 9 combine the two-loop complexity. The proofs of the theorems can be found in Appendix B.1.

---

**Algorithm 2** Extragradient Sliding for VIs

---

1: **Input:** $x^0 \in \mathbb{R}^d$
2: **Parameters:** $\eta, \theta, \alpha > 0, K \in \{1, 2, \ldots\}$
3: **for** $k = 0, 1, 2, \ldots, K-1$ **do**
4:     Find $u^k \approx \tilde{u}^k$ where $\tilde{u}^k$ is a solution for

$$\text{Find } \tilde{u}^k \in \mathbb{R}^d \ : \ B_\theta^k(\tilde{u}^k) = 0 \text{ with } B_\theta^k(x) := P(x^k) + Q(x) + \frac{1}{\theta}(x - x^k)$$

5:     $x^{k+1} = x^k + \eta\alpha(u^k - x^k) - \eta R(u^k)$
6: **end for**
7: **Output:** $x^K$

---

**Theorem 7.** *Consider Algorithm 2 for Problem (10) under Assumptions 7–9, with the following tuning:*

$$\theta = \tfrac{1}{2L_p}, \quad \eta = \min\left\{ \tfrac{1}{4\mu}, \tfrac{1}{4L_p} \right\}, \quad \alpha = 2\mu.$$

*Assume that $u^k$ (line 4) satisfies*

$$\|B_\theta^k(u^k)\|^2 \leq \tfrac{L_p^2}{3}\|x^k - \tilde{u}^k\|^2. \tag{12}$$

*Then, for any*

$$K \geq 2\max\left\{1, \tfrac{L_p}{\mu}\right\} \log \frac{\|x^0 - x^*\|^2}{\varepsilon}, \tag{13}$$

*we have the following estimate for the distance to the solution $x^*$:*

$$\|x^K - x^*\|^2 \leq \varepsilon. \tag{14}$$

The proof is given in Appendix B.1.

## 4.2 Solving the auxiliary problem

As for Algorithm 1, we need an auxiliary solver for the subproblem in line 4, which ensures that (12) is met. One can observe that $B_\theta^k(x)$ is $(2L_p + L_q)$-Lipschitz and monotone. For this type of problem, the authors in [49] proposes an approach that guarantees convergence on $\|B_\theta^k(u^k)\|^2$.

**Theorem 8** ([49] Corollary 2)**.** *There exists a certain algorithm that, applied to the subproblem (12) with starting point $x^k$, returns $u^k$ satisfying*

$$\|B_\theta^k(u^k)\|^2 \leq \frac{D^2 \cdot \max\{L_p^2, L_q^2\}\|x^k - \tilde{u}^k\|^2}{T^2}, \tag{15}$$

*where $D > 0$ is some universal numerical constant (independent of $L_p, L_q, T$ etc) and $T$ is the number of calls of the operator $Q$.*

## 4.3 Complexity of the optimal gradient sliding

Leveraging Theorems 7 and 8 while following the same reasoning as in Section 2.2, we obtain the following convergence (inner plus outer loops) for Algorithm 2.

**Theorem 9.** *Let Assumptions 7 to 9 be satisfied with $\mu \leq L_p \leq L_q$. Then, Algorithm 2 requires*

$$\mathcal{O}\left(\tfrac{L_q}{\mu} \log \tfrac{1}{\varepsilon}\right) \quad \text{calls of } Q(x) \quad \text{and} \quad \mathcal{O}\left(\tfrac{L_q}{\mu} \log \tfrac{1}{\varepsilon}\right) \quad \text{calls of } P(x)$$

*to find an $\varepsilon$-solution of problem (10).*

We also consider the case of monotone VIs (Assumption 7 with $\mu = 0$). For this we modify Algorithm 2 as in Appendix B.2 and obtain the following convergence result.

**Theorem 10.** *Let Assumption 7 (with $\mu = 0$), 8, 9 be satisfied and $L_p \leq L_q$. Then, Algorithm 4, described in Appendix B.2, requires*

$$\mathcal{O}\left(\tfrac{L_q}{\varepsilon}\|x^0 - x^*\|^2\right) \quad \text{calls of } Q(x) \quad \text{and} \quad \mathcal{O}\left(\tfrac{L_p}{\varepsilon}\|x^0 - x^*\|^2\right) \quad \text{calls of } P(x)$$

*to find an $\varepsilon$-solution of the problem (10). Here $\varepsilon$-solution is measured by the value of the gap function.*

## 4.4 Application to distributed saddle-point problem under similarity

We apply now Algorithm 2 to solve a distributed saddle-point problem under statistical similarity, as introduced in [9]:

$$\min_{y \in \mathbb{R}^{d_y}} \max_{z \in \mathbb{R}^{d_z}} r(y, z) := \tfrac{1}{n} \sum_{i=1}^{n} f_i(y, z). \tag{16}$$

**Assumption 10.** *Each $f_i(y, z) \colon \mathbb{R}^{d_y} \times \mathbb{R}^{d_z} \to \mathbb{R}$ is convex-concave and $L$-smooth on $\mathbb{R}^{d_y} \times \mathbb{R}^{d_z}$.*

**Assumption 11.** *$r(z, y)$ is $\mu$-strongly convex (first argument)–$\mu$-strongly concave (second argument).*

**Assumption 12.** *$f_1(y, z), \ldots, f_n(y, z)$ are $\delta$-related: for all $i \neq j$ and for all $y \in \mathbb{R}^{d_y}$ and $z \in \mathbb{R}^{d_z}$,*

$$\|\nabla_{yy}^2 f_i(y, z) - \nabla_{yy}^2 f_j(y, z)\| \leq \delta, \|\nabla_{yz}^2 f_i(y, z) - \nabla_{yz}^2 f_j(y, z)\| \leq \delta, \|\nabla_{zz}^2 f_i(y, z) - \nabla_{zz}^2 f_j(y, z)\| \leq \delta.$$

Casting (16) into the VI formulation (10), by taking $Q(x) = Q(y, z) = [\nabla_y f_1(y, z), -\nabla_z f_1(y, z)]$ and $P(x) = P(y, z) = [\nabla_y[r - f_1](y, z), -\nabla_z[r - f_1](y, z)]$, we have that $Q$ is monotone and $L$-Lipschitz, $P$ is $\delta$-Lipschitz, $R$ is $\mu$-strongly monotone. Therefore, we can apply Theorems 9 and 10 and obtain the following convergence results for Algorithm 2 applied to (16).

**Theorem 11.** *Let Assumptions 10 to 12 be satisfied with $\mu \le \delta \le L$. Then, to find $\varepsilon$-solution of the distributed saddle problem* (16)*, Algorithm 2 requires*

$$\mathcal{O}\left(\frac{\delta}{\mu} \log \frac{1}{\varepsilon}\right) \text{ communication rounds and } \mathcal{O}\left(\frac{L}{\mu} \log \frac{1}{\varepsilon}\right) \text{ local gradient computations.}$$

**Theorem 12.** *Let Assumptions 10, 11 (with $\mu = 0$), 12 be satisfied and $\delta \le L$. Then, to find $\varepsilon$-solution of the distributed saddle problem* (16)*, Algorithm 2 requires*

$$\mathcal{O}\left(\frac{\delta}{\varepsilon}\|x^0 - x^*\|^2\right) \text{ communication rounds and } \mathcal{O}\left(\frac{L}{\varepsilon}\|x^0 - x^*\|^2\right) \text{ local computations.}$$

Our communication estimates are optimal, as in [9], but our algorithm also achieve optimal local complexity (see Table 1).

# 5 Experiments

## 5.1 Minimization

We consider the Ridge Regression problem

$$\min_{w \in \mathbb{R}^d} \left[ f(w) := \frac{1}{2N} \sum_{i=1}^{N} (w^T x_i - y_i)^2 + \frac{\lambda}{2} \|w\|^2 \right], \tag{17}$$

where $w$ is the vector of weights of the model, $\{x_i, y_i\}_{i=1}^{N}$ is the training dataset, and $\lambda > 0$ is the regularization parameter.

We consider a network with 25 workers (simulated on a single-CPU machine), and use two types of datasets, namely: synthetic and real data. Synthetic data permit to control the similarity constant $\delta$. To do so, we generate data on the server, say $\{\hat{x}_i, \hat{y}_i\}_{i=1}^{n=100}$. Data on the workers are generated by adding unbiased Gaussian noise to the server data. The lower the variance of this noise, the more similar the data, and thus the smaller $\delta$. For simulations with real data, we considered the LIBSVM library [11] and give each agent a full dataset. Then, each device selects at random a part of size $m$ from the full dataset. In Section C.2 we explain how the parameters $L$ and $\delta$ are estimated. For the synthetic dataset we choose the noise level and the regularization parameter such that $L/\delta = 200$ and $L/\lambda = 10^5$. For the real datasets the regularization parameter is chosen such that $L/\lambda = 10^6$. See Table 2 for all values of $L, \delta, \mu$ and $m$.

For comparison, we use distributed version of Accelerated Gradient Descent (AcGD) [37] as the basic, classical and optimal method for minimization problems without additional similarity assumptions, as well as state-of-the-art schemes for the similarity condition: DANE [42], DANE-HB [50], SPAG [21] and AccSONATA [46]. The settings of the methods are made as described in the original papers. For algorithms that assume an absolutely accurate solution of local problems (DANE, SPAG, AccSONATA), we use AcGD with an accuracy of $10^{-12}$ as a subsolver.

Results are summarized in Figure 1–the first two figures from the top left correspond to synthetic data while the other 6 on real data.

The results show that our method outperforms almost all methods in terms of communication (only in one experiment SPAG is slightly faster). In terms of local iterations, our method is slightly inferior to AcGD, but superior to all other methods.

In Appendix C.1, one can find a comparison of our method with a competitor for distributed saddle point problems under similarity assumption.

# Acknowledgments

The research in Sections 1–3 was supported by Russian Science Foundation (project No. 21-71-30005). The work in Section 4 was prepared within the framework of the HSE University Basic Reaearch Programm. The research of A. Scutari was partially supported by the ONR Grant N. N00014-21-1-2673.

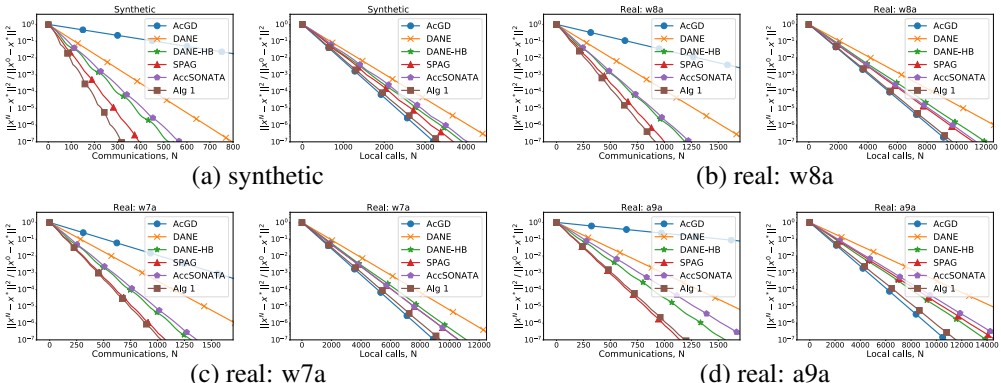

|  | (a) synthetic |  | (b) real: w8a |
|  | (c) real: w7a |  | (d) real: a9a |

Figure 1: Ridge regression problem (17): Comparison of state-of-the-art methods, under similarity; synthetic data (a) and real data (b,c,d). Distance from optimality vs. number of communications (first/third panel from the left in the both lines) and vs. number of local iterations (second/fourth panel from the left in the both lines).

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
