# APPENDIX

## Contents

# A  Proofs for Section 2

In this section we present a proof of the convergence of Algorithm 1 in the strongly convex case – Section A.1. We also present a modification of Algorithm 1 for the convex case, as well as a proof of its convergence – Section A.2.

## A.1  Strongly convex case

Here we prove Theorem 1. First, we need the following lemmas:

**Lemma 1.** *Consider Algorithm 1. Let $\theta$ be defined as in Theorem 1: $\theta = \frac{1}{2L_p}$. Then, under Assumptions 1-3, the following inequality holds for all $\bar{x} \in \mathbb{R}^d$*

$$
\begin{aligned}
2\langle \bar{x} - x_g^k, \nabla r(x_f^{k+1}) \rangle \leq & 2\left[ r(\bar{x}) - r(x_f^{k+1}) \right] - \mu \|x_f^{k+1} - \bar{x}\|^2 - \theta \|\nabla r(x_f^{k+1})\|^2 \\
& + 3\theta \left( \|\nabla A_\theta^k(x_f^{k+1})\|^2 - \frac{L_p^2}{3} \|x_g^k - \arg\min_{x \in \mathbb{R}^d} A_\theta^k(x)\|^2 \right).
\end{aligned} \tag{18}
$$

*Proof.* Using $\mu$-strong convexity of $r(x)$, we get

$$
\begin{aligned}
2\langle \bar{x} - x_g^k, \nabla r(x_f^{k+1}) \rangle = & 2\langle \bar{x} - x_f^{k+1}, \nabla r(x_f^{k+1}) \rangle + 2\langle x_f^{k+1} - x_g^k, \nabla r(x_f^{k+1}) \rangle \\
\leq & 2\left[ r(\bar{x}) - r(x_f^{k+1}) \right] - \mu \|x_f^{k+1} - \bar{x}\|^2 + 2\langle x_f^{k+1} - x_g^k, \nabla r(x_f^{k+1}) \rangle \\
= & 2\left[ r(\bar{x}) - r(x_f^{k+1}) \right] - \mu \|x_f^{k+1} - \bar{x}\|^2 + 2\theta\langle \theta^{-1}(x_f^{k+1} - x_g^k), \nabla r(x_f^{k+1}) \rangle \\
= & 2\left[ r(\bar{x}) - r(x_f^{k+1}) \right] - \mu \|x_f^{k+1} - \bar{x}\|^2 \\
& - \frac{1}{\theta} \|x_f^{k+1} - x_g^k\|^2 - \theta \|\nabla r(x_f^{k+1})\|^2 \\
& + \theta \|\theta^{-1}(x_f^{k+1} - x_g^k) + \nabla r(x_f^{k+1})\|^2.
\end{aligned}
$$

The definition of $A_\theta^k(x)$ and $L_p$-Lipschitzness of $\nabla p$ (Assumption 3) give

$$
\begin{aligned}
2\langle \bar{x} - x_g^k, \nabla r(x_f^{k+1}) \rangle \leq & 2\left[ r(\bar{x}) - r(x_f^{k+1}) \right] - \mu \|x_f^{k+1} - \bar{x}\|^2 - \frac{1}{\theta} \|x_f^{k+1} - x_g^k\|^2 - \theta \|\nabla r(x_f^{k+1})\|^2 \\
& + \theta \|\nabla A_\theta^k(x_f^{k+1}) + \nabla p(x_f^{k+1}) - \nabla p(x_g^k)\|^2 \\
\leq & 2\left[ r(\bar{x}) - r(x_f^{k+1}) \right] - \mu \|x_f^{k+1} - \bar{x}\|^2 - \frac{1}{\theta} \|x_f^{k+1} - x_g^k\|^2 - \theta \|\nabla r(x_f^{k+1})\|^2 \\
& + 2\theta \|\nabla A_\theta^k(x_f^{k+1})\|^2 + 2\theta L_p^2 \|x_f^{k+1} - x_g^k\|^2 \\
= & 2\left[ r(\bar{x}) - r(x_f^{k+1}) \right] - \mu \|x_f^{k+1} - \bar{x}\|^2 - \frac{1}{\theta} \left( 1 - 2\theta^2 L_p^2 \right) \|x_f^{k+1} - x_g^k\|^2 \\
& - \theta \|\nabla r(x_f^{k+1})\|^2 + 2\theta \|\nabla A_\theta^k(x_f^{k+1})\|^2.
\end{aligned}
$$

With $\theta = \frac{1}{2L_p}$, we have

$$
\begin{aligned}
2\langle \bar{x} - x_g^k, \nabla r(x_f^{k+1}) \rangle \leq & 2\left[ r(\bar{x}) - r(x_f^{k+1}) \right] - \mu \|x_f^{k+1} - \bar{x}\|^2 - \frac{1}{2\theta} \|x_f^{k+1} - x_g^k\|^2 \\
& - \theta \|\nabla r(x_f^{k+1})\|^2 + 2\theta \|\nabla A_\theta^k(x_f^{k+1})\|^2 \\
= & 2\left[ r(\bar{x}) - r(x_f^{k+1}) \right] - \mu \|x_f^{k+1} - \bar{x}\|^2 - \frac{1}{4\theta} \|x_g^k - \arg\min_{x \in \mathbb{R}^d} A_\theta^k(x)\|^2 \\
& + \frac{1}{2\theta} \|x_f^{k+1} - \arg\min_{x \in \mathbb{R}^d} A_\theta^k(x)\|^2 - \theta \|\nabla r(x_f^{k+1})\|^2 + 2\theta \|\nabla A_\theta^k(x_f^{k+1})\|^2.
\end{aligned}
$$

One can observe that $A_\theta^k(x)$ is $\frac{1}{\theta}$-strongly convex. Hence,

$$
\begin{aligned}
2\langle \bar{x} - x_g^k, \nabla r(x_f^{k+1})\rangle \leq &2\left[r(\bar{x}) - r(x_f^{k+1})\right] - \mu\|x_f^{k+1} - \bar{x}\|^2 - \frac{1}{4\theta}\|x_g^k - \arg\min_{x\in\mathbb{R}^d} A_\theta^k(x)\|^2 \\
&+ \frac{\theta}{2}\|\nabla A_\theta^k(x_f^{k+1})\|^2 - \theta\|\nabla r(x_f^{k+1})\|^2 + 2\theta\|\nabla A_\theta^k(x_f^{k+1})\|^2 \\
\leq &2\left[r(\bar{x}) - r(x_f^{k+1})\right] - \mu\|x_f^{k+1} - \bar{x}\|^2 - \frac{1}{4\theta}\|x_g^k - \arg\min_{x\in\mathbb{R}^d} A_\theta^k(x)\|^2 \\
&+ 3\theta\|\nabla A_\theta^k(x_f^{k+1})\|^2 - \theta\|\nabla r(x_f^{k+1})\|^2 \\
= &2\left[r(\bar{x}) - r(x_f^{k+1})\right] - \mu\|x_f^{k+1} - \bar{x}\|^2 - \theta\|\nabla r(x_f^{k+1})\|^2 \\
&+ 3\theta\left(\|\nabla A_\theta^k(x_f^{k+1})\|^2 - \frac{1}{12\theta^2}\|x_g^k - \arg\min_{x\in\mathbb{R}^d} A_\theta^k(x)\|^2\right) \\
= &2\left[r(\bar{x}) - r(x_f^{k+1})\right] - \mu\|x_f^{k+1} - \bar{x}\|^2 - \theta\|\nabla r(x_f^{k+1})\|^2 \\
&+ 3\theta\left(\|\nabla A_\theta^k(x_f^{k+1})\|^2 - \frac{L_p^2}{3}\|x_g^k - \arg\min_{x\in\mathbb{R}^d} A_\theta^k(x)\|^2\right).
\end{aligned}
$$

This completes the proof of Lemma. $\square$

**Lemma 2.** *Consider Algorithm 1 for Problem 1 under Assumptions 1-3, with the following tuning:*

$$
\tau = \min\left\{1, \frac{\sqrt{\mu}}{2\sqrt{L_p}}\right\}, \quad \theta = \frac{1}{2L_p}, \quad \eta = \min\left\{\frac{1}{2\mu}, \frac{1}{2\sqrt{\mu L_p}}\right\}, \quad \alpha = \mu, \tag{19}
$$

*and let $x_f^{k+1}$ in line 5 satisfy*

$$
\|\nabla A_\theta^k(x_f^{k+1})\|^2 \leq \frac{L_p^2}{3}\|x_g^k - \arg\min_{x\in\mathbb{R}^d} A_\theta^k(x)\|^2. \tag{20}
$$

*Then, the following inequality holds:*

$$
\frac{1}{\eta}\|x^{k+1} - x^*\|^2 + \frac{2}{\tau}\left[r(x_f^{k+1}) - r(x^*)\right] \leq (1-\rho)\left[\frac{1}{\eta}\|x^k - x^*\|^2 + \frac{2}{\tau}\left[r(x_f^k) - r(x^*)\right]\right], \tag{21}
$$

*where*

$$
\rho := \frac{1}{2}\min\left\{1, \sqrt{\frac{\mu}{L_p}}\right\}. \tag{22}
$$

*Proof.* Using line 6 of Algorithm 1, we get

$$
\begin{aligned}
\frac{1}{\eta}\|x^{k+1} - x^*\|^2 = &\frac{1}{\eta}\|x^k - x^*\|^2 + \frac{2}{\eta}\langle x^{k+1} - x^k, x^k - x^*\rangle + \frac{1}{\eta}\|x^{k+1} - x^k\|^2 \\
= &\frac{1}{\eta}\|x^k - x^*\|^2 + 2\alpha\langle x_f^{k+1} - x^k, x^k - x^*\rangle \\
&- 2\langle\nabla r(x_f^{k+1}), x^k - x^*\rangle + \frac{1}{\eta}\|x^{k+1} - x^k\|^2 \\
= &\frac{1}{\eta}\|x^k - x^*\|^2 + \alpha\|x_f^{k+1} - x^*\|^2 - \alpha\|x^k - x^*\|^2 - \alpha\|x_f^{k+1} - x^k\|^2 \\
&- 2\langle\nabla r(x_f^{k+1}), x^k - x^*\rangle + \frac{1}{\eta}\|x^{k+1} - x^k\|^2.
\end{aligned}
$$

Line 4 of Algorithm 1 gives

$$
\begin{aligned}
\frac{1}{\eta}\|x^{k+1} - x^*\|^2 = &\left(\frac{1}{\eta} - \alpha\right)\|x^k - x^*\|^2 + \alpha\|x_f^{k+1} - x^*\|^2 + \frac{1}{\eta}\|x^{k+1} - x^k\|^2 - \alpha\|x_f^{k+1} - x^k\|^2 \\
&+ 2\langle\nabla r(x_f^{k+1}), x^* - x_g^k\rangle + \frac{2(1-\tau)}{\tau}\langle\nabla r(x_f^{k+1}), x_f^k - x_g^k\rangle.
\end{aligned}
$$

Using (18) with $\bar{x} = x^*$ and $\bar{x} = x_f^k$, we get

$$\frac{1}{\eta}\|x^{k+1} - x^*\|^2 \leq \left(\frac{1}{\eta} - \alpha\right)\|x^k - x^*\|^2 + \alpha\|x_f^{k+1} - x^*\|^2 + \frac{1}{\eta}\|x^{k+1} - x^k\|^2 - \alpha\|x_f^{k+1} - x^k\|^2$$

$$+ 2\left[r(x^*) - r(x_f^{k+1})\right] - \mu\|x_f^{k+1} - x^*\|^2 + \frac{2(1-\tau)}{\tau}\left[r(x_f^k) - r(x_f^{k+1})\right]$$

$$- \frac{1}{2\tau L_p}\|\nabla r(x_f^{k+1})\|^2 + \frac{3\theta}{\tau}\left(\|\nabla A_\theta^k(x_f^{k+1})\|^2 - \frac{L_p^2}{3}\|x_g^k - \arg\min_{x \in \mathbb{R}^d} A_\theta^k(x)\|^2\right)$$

$$= \left(\frac{1}{\eta} - \alpha\right)\|x^k - x^*\|^2 + (\alpha - \mu)\|x_f^{k+1} - x^*\|^2 - \alpha\|x_f^{k+1} - x^k\|^2$$

$$+ \frac{1}{\eta}\|\eta\alpha(x_f^{k+1} - x^k) - \eta\nabla r(x_f^{k+1})\|^2 - \frac{1}{2\tau L_p}\|\nabla r(x_f^{k+1})\|^2$$

$$+ \frac{2(1-\tau)}{\tau}\left[r(x_f^k) - r(x^*)\right] - \frac{2}{\tau}\left[r(x_f^{k+1}) - r(x^*)\right]$$

$$+ \frac{3\theta}{\tau}\left(\|\nabla A_\theta^k(x_f^{k+1})\|^2 - \frac{L_p^2}{3}\|x_g^k - \arg\min_{x \in \mathbb{R}^d} A_\theta^k(x)\|^2\right)$$

$$\leq \left(\frac{1}{\eta} - \alpha\right)\|x^k - x^*\|^2 + (\alpha - \mu)\|x_f^{k+1} - x^*\|^2$$

$$+ \alpha(2\eta\alpha - 1)\|x_f^{k+1} - x^k\|^2 + \left(2\eta - \frac{1}{2\tau L_p}\right)\|\nabla r(x_f^{k+1})\|^2$$

$$+ \frac{2(1-\tau)}{\tau}\left[r(x_f^k) - r(x^*)\right] - \frac{2}{\tau}\left[r(x_f^{k+1}) - r(x^*)\right]$$

$$+ \frac{3\theta}{\tau}\left(\|\nabla A_\theta^k(x_f^{k+1})\|^2 - \frac{L_p^2}{3}\|x_g^k - \arg\min_{x \in \mathbb{R}^d} A_\theta^k(x)\|^2\right).$$

The choice of $\alpha, \eta, \tau$ defined by (19) gives

$$\frac{1}{\eta}\|x^{k+1} - x^*\|^2 \leq \left(\frac{1}{\eta} - \alpha\right)\|x^k - x^*\|^2 + \frac{2(1-\tau)}{\tau}\left[r(x_f^k) - r(x^*)\right] - \frac{2}{\tau}\left[r(x_f^{k+1}) - r(x^*)\right]$$

$$+ \frac{3\theta}{\tau}\left(\|\nabla A_\theta^k(x_f^{k+1})\|^2 - \frac{L_p^2}{3}\|x_g^k - \arg\min_{x \in \mathbb{R}^d} A_\theta^k(x)\|^2\right).$$

With (20), we have

$$\frac{1}{\eta}\|x^{k+1} - x^*\|^2 + \frac{2}{\tau}\left[r(x_f^{k+1}) - r(x^*)\right] \leq \left(\frac{1}{\eta} - \alpha\right)\|x^k - x^*\|^2 + \frac{2(1-\tau)}{\tau}\left[r(x_f^k) - r(x^*)\right]$$

$$\leq (1 - \rho)\left[\frac{1}{\eta}\|x^k - x^*\|^2 + \frac{2}{\tau}\left[r(x_f^k) - r(x^*)\right]\right],$$

where $\rho$ is defined by (22). $\qquad\square$

To prove Theorem 1, it is sufficient to run the recursion (21):

$$\|x^K - x^*\|^2 \leq (1 - \rho)^K\left[\|x^0 - x^*\|^2 + \frac{2\eta}{\tau}\left[r(x^0) - r(x^*)\right]\right] = C(1 - \rho)^K,$$

where $C$ is defined as

$$C = \|x^0 - x^*\|^2 + \frac{2\eta}{\tau}\left[r(x^0) - r(x^*)\right].$$

Hence, choosing number of iterations $K$ given by (5) yields

$$\|x^K - x^*\|^2 \leq \varepsilon.$$

## A.2 Convex case

The next Algorithm 3 is an adaptation of Algorithm 1 for the convex case. In particular, time-varying $\tau_{k+1}$ and $\eta_{k+1}$ are used instead of the momentum $\alpha$.

---

**Algorithm 3** Accelerated Extragradient (modification for convex case)

1: **Input:** $x^0 = x_f^0 \in \mathbb{R}^d$
2: **Parameters:** $K \in \{1, 2, \ldots\}, \{\tau_k\}_{k=1}^K \subset (0, 1], \{\eta_k\}_{k=1}^K \subset \mathbb{R}_+, \theta > 0$
3: **for** $k = 0, 1, 2, \ldots, K - 1$ **do**
4: $\quad x_g^k = \tau_{k+1} x^k + (1 - \tau_{k+1}) x_f^k$
5: $\quad x_f^{k+1} \approx \arg \min_{x \in \mathbb{R}^d} \left[ A_\theta^k(x) := p(x_g^k) + \langle \nabla p(x_g^k), x - x_g^k \rangle + \frac{1}{2\theta} \| x - x_g^k \|^2 + q(x) \right]$
6: $\quad x^{k+1} = x^k - \eta_{k+1} \nabla r(x_f^{k+1})$
7: **end for**
8: **Output:** $x_f^K$

---

**Lemma 3.** *Consider Algorithm 3 for Problem 1 under Assumptions 1 ($\mu = 0$)–3, with the following tuning:*

$$\tau_k = \frac{2}{k+1}, \quad \theta = \frac{1}{2L_p}, \quad \eta_k = \frac{1}{2\tau_k L_p}, \tag{23}$$

*and let $x_f^{k+1}$ in line 5 satisfy*

$$\|\nabla A_\theta^k(x_f^{k+1})\|^2 \leq \frac{L_p^2}{3} \|x_g^k - \arg\min_{x \in \mathbb{R}^d} A_\theta^k(x)\|^2. \tag{24}$$

*Then, the following inequality holds:*

$$r(x_f^k) - r(x^*) \leq \frac{4L_p}{(k+1)^2} \|x^0 - x^*\|^2. \tag{25}$$

*Proof.* We start from line 6 of Algorithm 3 and get

$$\frac{1}{\eta_{k+1}} \|x^{k+1} - x^*\|^2 = \frac{1}{\eta_{k+1}} \|x^k - x^*\|^2 + \frac{2}{\eta_{k+1}} \langle x^{k+1} - x^k, x^k - x^* \rangle + \frac{1}{\eta_{k+1}} \|x^{k+1} - x^k\|^2$$

$$= \frac{1}{\eta_{k+1}} \|x^k - x^*\|^2 - 2\langle \nabla r(x_f^{k+1}), x^k - x^* \rangle + \frac{1}{\eta_{k+1}} \|x^{k+1} - x^k\|^2.$$

Line 4 of Algorithm 3 gives

$$\frac{1}{\eta_{k+1}} \|x^{k+1} - x^*\|^2 = \frac{1}{\eta_{k+1}} \|x^k - x^*\|^2 + \frac{1}{\eta_{k+1}} \|x^{k+1} - x^k\|^2$$

$$+ 2\langle \nabla r(x_f^{k+1}), x^* - x_g^k \rangle + \frac{2(1 - \tau_{k+1})}{\tau_{k+1}} \langle \nabla r(x_f^{k+1}), x_f^k - x_g^k \rangle.$$

Using (18) with $\mu = 0$, $\bar{x} = x^*$ and $\bar{x} = x_f^k$, we get

$$\frac{1}{\eta_{k+1}} \|x^{k+1} - x^*\|^2 \leq \frac{1}{\eta_{k+1}} \|x^k - x^*\|^2 + \frac{1}{\eta_{k+1}} \|x^{k+1} - x^k\|^2 + 2\left[ r(x^*) - r(x_f^{k+1}) \right]$$

$$+ \frac{2(1 - \tau_{k+1})}{\tau_{k+1}} \left[ r(x_f^k) - r(x_f^{k+1}) \right] - \frac{1}{2\tau_{k+1} L_p} \|\nabla r(x_f^{k+1})\|^2$$

$$+ \frac{3\theta}{\tau_{k+1}} \left( \|\nabla A_\theta^k(x_f^{k+1})\|^2 - \frac{L_p^2}{3} \|x_g^k - \arg\min_{x \in \mathbb{R}^d} A_\theta^k(x)\|^2 \right)$$

$$= \frac{1}{\eta_{k+1}} \|x^k - x^*\|^2 + \frac{1}{\eta_{k+1}} \|\eta_k \nabla r(x_f^{k+1})\|^2 - \frac{1}{2\tau_{k+1} L_p} \|\nabla r(x_f^{k+1})\|^2$$

$$+ \frac{2(1 - \tau_{k+1})}{\tau_{k+1}} \left[ r(x_f^k) - r(x^*) \right] - \frac{2}{\tau_{k+1}} \left[ r(x_f^{k+1}) - r(x^*) \right]$$

$$+ \frac{3\theta}{\tau_{k+1}} \left( \|\nabla A_\theta^k(x_f^{k+1})\|^2 - \frac{L_p^2}{3} \|x_g^k - \operatorname*{arg\,min}_{x \in \mathbb{R}^d} A_\theta^k(x)\|^2 \right)$$

$$= \frac{1}{\eta_{k+1}} \|x^k - x^*\|^2 + \left( \eta_{k+1} - \frac{1}{2\tau_{k+1} L_p} \right) \|\nabla r(x_f^{k+1})\|^2$$

$$+ \frac{2(1 - \tau_{k+1})}{\tau_{k+1}} \left[ r(x_f^k) - r(x^*) \right] - \frac{2}{\tau_{k+1}} \left[ r(x_f^{k+1}) - r(x^*) \right]$$

$$+ \frac{3\theta}{\tau_{k+1}} \left( \|\nabla A_\theta^k(x_f^{k+1})\|^2 - \frac{L_p^2}{3} \|x_g^k - \operatorname*{arg\,min}_{x \in \mathbb{R}^d} A_\theta^k(x)\|^2 \right).$$

The choice of $\eta_k$ defined by (23) gives

$$\|x^{k+1} - x^*\|^2 \leq \|x^k - x^*\|^2 + \frac{1 - \tau_{k+1}}{\tau_{k+1}^2 L_p} \left[ r(x_f^k) - r(x^*) \right] - \frac{1}{\tau_{k+1}^2 L_p} \left[ r(x_f^{k+1}) - r(x^*) \right]$$

$$+ \frac{3\theta}{2\tau_{k+1}^2 L_p} \left( \|\nabla A_\theta^k(x_f^{k+1})\|^2 - \frac{L_p^2}{3} \|x_g^k - \operatorname*{arg\,min}_{x \in \mathbb{R}^d} A_\theta^k(x)\|^2 \right).$$

With (24), we have

$$\|x^{k+1} - x^*\|^2 + \frac{1}{\tau_{k+1}^2 L_p} \left[ r(x_f^{k+1}) - r(x^*) \right] \leq \|x^k - x^*\|^2 + \frac{1 - \tau_{k+1}}{\tau_{k+1}^2 L_p} \left[ r(x_f^k) - r(x^*) \right]. \quad (26)$$

Let us define $\Psi_k$:

$$\Psi_k := \|x^k - x^*\|^2 + \frac{1}{\tau_k^2 L_p} \left[ r(x_f^k) - r(x^*) \right].$$

Using (26), $\Psi_k$ defined above and $\tau_k$ defined by (23) we get:

$$\frac{1}{\tau_{k+1} L_p} \left[ r(x_f^{k+1}) - r(x^*) \right] \leq \Psi_{k+1}$$

$$\leq \|x^k - x^*\|^2 + \frac{1 - \tau_{k+1}}{\tau_{k+1}^2 L_p} \left[ r(x_f^k) - r(x^*) \right]$$

$$= \|x^k - x^*\|^2 + \frac{(k+2)^2 - 2(k+2)}{4L_p} \left[ r(x_f^k) - r(x^*) \right]$$

$$= \|x^k - x^*\|^2 + \frac{(k+2)k}{4L_p} \left[ r(x_f^k) - r(x^*) \right]$$

$$\leq \|x^k - x^*\|^2 + \frac{(k+1)^2}{4L_p} \left[ r(x_f^k) - r(x^*) \right]$$

$$= \|x^k - x^*\|^2 + \frac{1}{\tau_k^2 L_p} \left[ r(x_f^k) - r(x^*) \right] = \Psi_k.$$

Next, we apply the previous inequality

$$\frac{1}{\tau_{k+1} L_p} \left[ r(x_f^{k+1}) - r(x^*) \right] \leq \Psi_{k+1} \leq \Psi_k \leq \ldots \leq \Psi_1 \leq \|x^0 - x^*\|^2 + \frac{1 - \tau_1}{\tau_1^2 L_p} \left[ r(x_f^1) - r(x^*) \right].$$

With $\tau_1 = 1$, we have

$$\frac{1}{\tau_{k+1} L_p} \left[ r(x_f^{k+1}) - r(x^*) \right] \leq \|x^0 - x^*\|^2.$$

Finally, again with the choice of $\tau_k$ defined by (23), we get (25). $\qquad \square$

Using (25), we get

$$r(x_f^K) - r(x^*) \leq \varepsilon$$

after

$$T = \sqrt{\frac{4L_p}{\varepsilon}} \|x^0 - x^*\|$$

iterations of Algorithm 3. This is what Theorem 4 is about.

## B   Proofs for Section 4

In this section we present a proof of the convergence of Algorithm 2 in the strongly monotone case – Section B.1. We also present a modification of Algorithm 2 for the monotone case, as well as a proof of its convergence – Section B.2.

### B.1   Strongly monotone case

Here we prove Theorem 7. First, we need the following lemmas:

**Lemma 4.** *Consider Algorithm 2. Let $\theta$ be defined as in Theorem 7: $\theta = \frac{1}{2L_p}$. Then, under Assumptions 7-9, the following inequality holds for all $\bar{x} \in \mathbb{R}^d$*

$$2\langle x^* - x^k, R(u^k)\rangle \leq -2\mu\|u^k - x^*\|^2 - \theta\|R(u^k)\|^2$$

$$+ 3\theta\left(\|B_\theta^k(u^k)\|^2 - \frac{L_p^2}{3}\|x^k - \tilde{u}^k\|^2\right). \tag{27}$$

*Proof.* Using property of the solution: $R(x^*) = 0$ and $\mu$-strong monotonicity of $R(x)$, we get

$$2\langle x^* - x^k, R(u^k)\rangle = 2\langle x^* - u^k, R(u^k)\rangle + 2\langle u^k - x^k, R(u^k)\rangle$$

$$\leq 2\langle x^* - u^k, R(u^k) - R(x^*)\rangle + 2\langle u^k - x^k, R(u^k)\rangle$$

$$\leq -2\mu\|u^k - x^*\|^2 + 2\langle u^k - x^k, R(u^k)\rangle$$

$$= -2\mu\|u^k - x^*\|^2 + 2\theta\langle \theta^{-1}(u^k - x^k), R(u^k)\rangle$$

$$= -2\mu\|u^k - x^*\|^2$$

$$- \frac{1}{\theta}\|u^k - x^k\|^2 - \theta\|R(u^k)\|^2 + \theta\|\theta^{-1}(u^k - x^k) + R(u^k)\|^2.$$

The definition of $B_\theta^k(x)$ and $L_p$-Lipschitzness of $P$ (Assumption 9) give

$$2\langle x^* - x^k, R(u^k)\rangle \leq -2\mu\|u^k - x^*\|^2 - \frac{1}{\theta}\|u^k - x^k\|^2 - \theta\|R(u^k)\|^2$$

$$+ \theta\|B_\theta^k(u^k) + P(u^k) - P(x^k)\|^2$$

$$\leq -2\mu\|u^k - x^*\|^2 - \frac{1}{\theta}\|u^k - x^k\|^2 - \theta\|R(u^k)\|^2$$

$$+ 2\theta\|B_\theta^k(u^k)\|^2 + 2\theta L_p^2\|u^k - x^k\|^2$$

$$= -2\mu\|u^k - x^*\|^2 - \frac{1}{\theta}\left(1 - 2\theta^2 L_p^2\right)\|u^k - x^k\|^2$$

$$- \theta\|R(u^k)\|^2 + 2\theta\|B_\theta^k(u^k)\|^2.$$

With $\theta = \frac{1}{2L_p}$, we have

$$2\langle x^* - x^k, R(u^k)\rangle \leq -2\mu\|u^k - x^*\|^2 - \frac{1}{2\theta}\|u^k - x^k\|^2 - \theta\|R(u^k)\|^2 + 2\theta\|B_\theta^k(u^k)\|^2$$

$$= -2\mu\|u^k - \bar{x}\|^2 - \frac{1}{4\theta}\|x^k - \tilde{u}^k\|^2$$

$$+ \frac{1}{2\theta}\|u^k - \tilde{u}^k\|^2 - \theta\|R(u^k)\|^2 + 2\theta\|B_\theta^k(u^k)\|^2.$$

One can observe that $B_\theta^k(x)$ is $\frac{1}{\theta}$-strongly monotone. It gives that

$$\frac{1}{\theta}\|x - y\|^2 \leq \langle B_\theta^k(x) - B_\theta^k(y); x - y\rangle \leq \|B_\theta^k(x) - B_\theta^k(y)\| \cdot \|x - y\|,$$

and with $B_\theta^k(\tilde{u}^k) = 0$ (since $\tilde{u}^k$ is the solution of line 4), we get

$$\frac{1}{\theta^2}\|u^k - \tilde{u}^k\|^2 \leq \|B_\theta^k(u^k) - B_\theta^k(\tilde{u}^k)\|^2 = \|B_\theta^k(u^k)\|^2.$$

Hence,

$$
\begin{aligned}
2\langle x^* - x^k, R(u^k)\rangle \leq & - 2\mu\|u^k - x^*\|^2 - \frac{1}{4\theta}\|x^k - \tilde{u}^k\|^2 \\
& + \frac{\theta}{2}\|B_\theta^k(u^k)\|^2 - \theta\|R(u^k)\|^2 + 2\theta\|B_\theta^k(u^k)\|^2 \\
\leq & - 2\mu\|u^k - \bar{x}\|^2 - \frac{1}{4\theta}\|x^k - \tilde{u}^k\|^2 \\
& + 3\theta\|B_\theta^k(u^k)\|^2 - \theta\|R(u^k)\|^2 \\
= & - 2\mu\|u^k - \bar{x}\|^2 - \theta\|R(u^k)\|^2 \\
& + 3\theta\left(\|B_\theta^k(u^k)\|^2 - \frac{1}{12\theta^2}\|x^k - \tilde{u}^k\|^2\right) \\
= & - 2\mu\|u^k - \bar{x}\|^2 - \theta\|R(u^k)\|^2 \\
& + 3\theta\left(\|B_\theta^k(u^k)\|^2 - \frac{L_p^2}{3}\|x^k - \tilde{u}^k\|^2\right).
\end{aligned}
$$

This completes the proof of Lemma. $\qquad\square$

**Lemma 5.** *Consider Algorithm 2 for Problem 10 under Assumptions 7-9, with the following tuning:*

$$
\theta = \frac{1}{2L_p}, \quad \eta = \min\left\{\frac{1}{4\mu}, \frac{1}{4L_p}\right\}, \quad \alpha = 2\mu, \tag{28}
$$

*and let $u^k$ in line 4 satisfies*

$$
\|B_\theta^k(u^k)\|^2 \leq \frac{L_p^2}{3}\|x^k - \tilde{u}^k\|^2. \tag{29}
$$

*Then, the following inequality holds:*

$$
\|x^{k+1} - x^*\|^2 \leq (1 - 2\mu\eta)^K \|x^0 - x^*\|^2. \tag{30}
$$

*Proof.* Using line 5 of Algorithm 2, we get

$$
\begin{aligned}
\|x^{k+1} - x^*\|^2 = & \|x^k - x^*\|^2 + 2\langle x^{k+1} - x^k, x^k - x^*\rangle + \|x^{k+1} - x^k\|^2 \\
= & \|x^k - x^*\|^2 + 2\eta\alpha\langle u^k - x^k, x^k - x^*\rangle \\
& - 2\eta\langle R(u^k), x^k - x^*\rangle + \|x^{k+1} - x^k\|^2 \\
= & \|x^k - x^*\|^2 + \eta\alpha\|u^k - x^*\|^2 - \eta\alpha\|x^k - x^*\|^2 - \eta\alpha\|u^k - x^k\|^2 \\
& - 2\eta\langle R(u^k), x^k - x^*\rangle + \|x^{k+1} - x^k\|^2.
\end{aligned}
$$

With (27), we get

$$
\begin{aligned}
\|x^{k+1} - x^*\|^2 \leq & (1 - \eta\alpha)\|x^k - x^*\|^2 + \eta\alpha\|u^k - x^*\|^2 + \|x^{k+1} - x^k\|^2 - \eta\alpha\|u^k - x^k\|^2 \\
& - 2\eta\mu\|u^k - x^*\|^2 - \eta\theta\|R(u^k)\|^2 \\
& + 3\eta\theta\left(\|B_\theta^k(u^k)\|^2 - \frac{L_p^2}{3}\|x^k - \tilde{u}^k\|^2\right) \\
= & (1 - \eta\alpha)\|x^k - x^*\|^2 + \|\eta\alpha(u^k - x^k) - \eta R(u^k)\|^2 - \eta\alpha\|u^k - x^k\|^2 \\
& - \eta(2\mu - \alpha)\|u^k - x^*\|^2 - \eta\theta\|R(u^k)\|^2 \\
& + 3\eta\theta\left(\|B_\theta^k(u^k)\|^2 - \frac{L_p^2}{3}\|x^k - \tilde{u}^k\|^2\right) \\
\leq & (1 - \eta\alpha)\|x^k - x^*\|^2 - \eta\alpha(1 - 2\eta\alpha)\|u^k - x^k\|^2 \\
& - \eta(2\mu - \alpha)\|u^k - x^*\|^2 - \eta(\theta - 2\eta)\|R(u^k)\|^2 \\
& + 3\eta\theta\left(\|B_\theta^k(u^k)\|^2 - \frac{L_p^2}{3}\|x^k - \tilde{u}^k\|^2\right).
\end{aligned}
$$

The choice of $\alpha, \eta, \theta$ defined by (28) gives

$$\|x^{k+1} - x^*\|^2 \leq (1 - 2\eta\mu) \|x^k - x^*\|^2 + 3\eta\theta \left( \|B_\theta^k(u^k)\|^2 - \frac{L_p^2}{3} \|x^k - \tilde{u}^k\|^2 \right).$$

Using (12) we get

$$\|x^{k+1} - x^*\|^2 \leq (1 - 2\eta\mu) \|x^k - x^*\|^2.$$

$\square$

To prove Theorem 7, it is sufficient to run the recursion (30):

$$\|x^K - x^*\|^2 \leq (1 - 2\eta\mu)^K \left[ \|x^0 - x^*\|^2 + \frac{2\eta}{\tau} \left[ r(x^0) - r(x^*) \right] \right] = C(1 - \rho)^K,$$

Hence, choosing number of iterations $K$ given by (13) yields

$$\|x^K - x^*\|^2 \leq \varepsilon.$$

## B.2   Monotone case

The next Algorithm 4 is an adaptation of Algorithm 2 for the monotone case. In particular, we remove the momentum $\alpha$.

---

**Algorithm 4** Extragradient (modification for monotone case)

---

1: **Input:** $x^0 \in \mathbb{R}^d$
2: **Parameters:** $\eta, \theta > 0, K \in \{1, 2, \ldots\}$
3: **for** $k = 0, 1, 2, \ldots, K - 1$ **do**
4:     Find $u^k \approx \tilde{u}^k$ where $\tilde{u}^k$ is a solution for
        Find $\tilde{u}^k \in \mathbb{R}^d \ : \ B_\theta^k(\tilde{u}^k) = 0$ with $B_\theta^k(x) := P(x^k) + Q(x) + \frac{1}{\theta}(x - x^k)$
5:     $x^{k+1} = x^k - \eta R(u^k)$
6: **end for**
7: **Output:** $x^K$

---

**Lemma 6.** *Consider Algorithm 4 for Problem 10 under Assumptions 7 ($\mu = 0$)-9 with the following tuning:*

$$\theta = \frac{1}{2L_p}, \quad \eta = \frac{1}{4L_p}, \tag{31}$$

*and let $u^k$ in line 4 satisfies*

$$\|B_\theta^k(u^k)\|^2 \leq \frac{L_p^2}{3} \|x^k - \tilde{u}^k\|^2. \tag{32}$$

*Then, the following inequality holds:*

$$\sup_{x \in \mathcal{C}} \langle R(x), \left( \frac{1}{K} \sum_{k=0}^{K-1} u^k \right) - x \rangle \leq \frac{2L_p \|x^0 - x\|^2}{K}. \tag{33}$$

**Remark 1.** *Here we do not take the maximum over the entire set $\mathbb{R}^d$ (as in the classical version for VIs [23]), but over $\mathcal{C}$ – a compact subset of $\mathbb{R}^d$. Thus, we can also consider unbounded sets in $\mathbb{R}^d$. This is permissible, since such a version of the criterion is valid if the solution $x^*$ lies in $\mathcal{C}$; for details see the work of [36].*

*Proof.* We start from line 5 of Algorithm 4 and get

$$\|x^{k+1} - x\|^2 = \|x^k - x\|^2 + 2\langle x^{k+1} - x^k, x^k - x \rangle + \|x^{k+1} - x^k\|^2$$
$$= \|x^k - x\|^2 - 2\eta\langle R(u^k), x^k - x \rangle + \|x^{k+1} - x^k\|^2.$$

Using (27) with $\mu = 0$, we get

$$
\begin{aligned}
\|x^{k+1} - x\|^2 \leq & \|x^k - x\|^2 + \|x^{k+1} - x^k\|^2 \\
& - 2\eta\langle R(u^k), u^k - x\rangle - \eta\theta\|R(u^k)\|^2 \\
& + 3\eta\theta\left(\|B_\theta^k(u^k)\|^2 - \frac{L_p^2}{3}\|x^k - \tilde{u}^k\|^2\right) \\
= & \|x^k - x\|^2 - 2\eta\langle R(u^k), u^k - x\rangle + \eta^2\|R(u^k)\|^2 - \eta\theta\|R(u^k)\|^2 \\
& + 3\eta\theta\left(\|B_\theta^k(u^k)\|^2 - \frac{L_p^2}{3}\|x^k - \tilde{u}^k\|^2\right) \\
\leq & \|x^k - x\|^2 - 2\eta\langle R(u^k), u^k - x\rangle - \eta(\theta - \eta)\|R(u^k)\|^2 \\
& + 3\eta\theta\left(\|B_\theta^k(u^k)\|^2 - \frac{L_p^2}{3}\|x^k - \tilde{u}^k\|^2\right).
\end{aligned}
$$

The choice of $\theta, \eta$ defined by (23) give

$$
\begin{aligned}
\|x^{k+1} - x\|^2 \leq & \|x^k - x\|^2 - 2\eta\langle R(u^k), u^k - x\rangle \\
& + 3\eta\theta\left(\|B_\theta^k(u^k)\|^2 - \frac{L_p^2}{3}\|x^k - \tilde{u}^k\|^2\right).
\end{aligned}
$$

With (32), we have

$$
\|x^{k+1} - x\|^2 \leq \|x^k - x\|^2 - 2\eta\langle R(u^k), u^k - x\rangle.
$$

Summing from $k = 0$ to $K - 1$, we obtain

$$
\begin{aligned}
\frac{1}{K}\sum_{k=0}^{K-1}\langle R(u^k), u^k - x\rangle &\leq \frac{\|x^0 - x\|^2 - \|x^K - x\|^2}{2\eta K} \\
&\leq \frac{\|x^0 - x\|^2}{2\eta K}.
\end{aligned}
$$

Monotonicity of $R$ gives

$$
\begin{aligned}
\left\langle R(x), \left(\frac{1}{K}\sum_{k=0}^{K-1}u^k\right) - x\right\rangle &= \frac{1}{K}\sum_{k=0}^{K-1}\langle R(x), u^k - x\rangle \\
&\leq \frac{1}{K}\sum_{k=0}^{K-1}\langle R(u^k), u^k - x\rangle \\
&\leq \frac{\|x^0 - x\|^2}{2\eta K} \leq \frac{2L_p\|x^0 - x\|^2}{K}.
\end{aligned}
$$

By taking the supremum over the set $\mathcal{C}$, we get

$$
\sup_{x\in\mathcal{C}}\left\langle R(x), \left(\frac{1}{K}\sum_{k=0}^{K-1}u^k\right) - x\right\rangle \leq \frac{2L_p\|x^0 - x\|^2}{K}.
$$

$\square$

Using (33), we get

$$
\sup_{x\in\mathcal{C}}\left\langle R(x), \left(\frac{1}{K}\sum_{k=0}^{K-1}u^k\right) - x\right\rangle \leq \varepsilon
$$

after

$$
T = \frac{2L_p}{\varepsilon}\|x^0 - x^*\|^2
$$

iterations of Algorithm 4. This is what Theorem 10 is about.

## C  Additional experiments

### C.1  Additional experiments with saddle point problems

Here we consider a modification of (17), the Robust Linear Regression, which leads to the following saddle-point formulation:

$$\min_w \max_{\|r_i\| \le R_r} \frac{1}{2N} \sum_{i=1}^{N} \left[ (w^T(x_i + r_i) - y_i)^2 - \beta\|r_i\|^2 \right] + \frac{\lambda}{2}\|w\|^2, \tag{34}$$

where $r_i$ is the so-called adversarial noise and $\beta > 0$ is the regularization associated with it; we set $\lambda = \beta = 0, 1$ and $R_r = 0, 05$. The network setting and data generation is the same as discussed in Section 5.1. We compare with the only existing method for SPPs under similarity, as proposed in [9]. Results are summarized in Figure 2, on synthetic and real data.

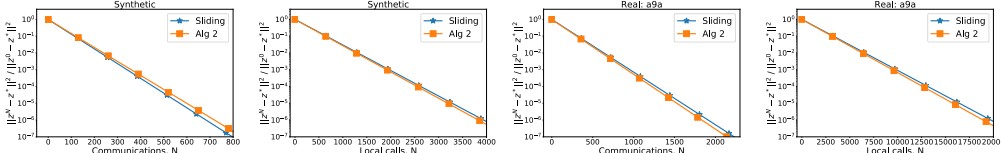

Figure 2: Robust Linear Regression (34), under similarity assumption: Proposed method vs. Gradient-Sliding; synthetic data (first two figures on the left) and real data (last to figures from the right). Distance from optimality vs. number of communications (first/third panel from the left) and vs. number of local iterations (second/fourth panel from the left).

It can be seen that our method compares favorably with [9] both on communication and gradient iterations.

### C.2  Experiment details

The numerical experiments are run on a machine with 8 Intel Core(TM) i7-9700KF 3.60GHz CPU cores with 64GB RAM. The methods are implemented in Python 3.7 using NumPy and SciPy.

In this section, we estimate the smoothness, strong convexity as well as the similarity parameters for objective (17). We denote the identity matrix as $I$ (with the sizes determined by the context). Given a set of data points $X = (x_1 \ldots x_N)^\top \in \mathbb{R}^{N \times d}$ and an associated set of labels $y = (y_1 \ldots y_N)^\top \in \mathbb{R}^N$, the Linear Regression problem (17) is

$$\min_{\|w\|} g(w) := \frac{1}{2N} \sum_{i=1}^{N} (w^T x_i - y_i)^2 + \frac{\lambda}{2}\|w\|^2.$$

Equivalently, $g(w)$ can be expressed as

$$g(w) = \frac{1}{2N}\|Xw - y\|^2 + \frac{\lambda}{2}\|w\|^2,$$

and its gradient writes as

$$\nabla g(w) = \frac{1}{N}\left(X^\top X w - X^\top y\right) \lambda w.$$

The Hessian of $g(w)$ is

$$\nabla^2 g(w) = \frac{1}{N} X^\top X + \lambda I.$$

We are now ready to estimate the spectrum of the Hessian

$$\|\nabla^2 g(w) v\| \le \frac{1}{N}\lambda_{\max}(X^\top X)\|v\| + \lambda\|v\|$$

$$\le \left(\frac{1}{N}\lambda_{\max}(X^\top X) + \lambda\right) \cdot \|v\| =: L_g\|v\|.$$

Therefore, we can estimate the Lipschitz constant of $\nabla g(w)$ as $L_g$. The same way we can estimate all $L_i$ and take final $L = \max(L_g, L_1, \ldots, L_n)$.

Let us discuss the bound on the similarity parameter. Given two datasets $\{X \in \mathbb{R}^{N \times d}, \; y \in \mathbb{R}^N\}$ and $\{\widetilde{X} \in \mathbb{R}^{\widetilde{N} \times d}, \; \widetilde{y} \in \mathbb{R}^{\widetilde{N}}\}$, we define

$$\widetilde{g}(w) = \frac{1}{2\widetilde{N}} \|\widetilde{X}w - \widetilde{y}\|^2 + \frac{\lambda}{2} \|w\|^2.$$

And then the similarity coefficient $\delta^{g, \widetilde{g}}$ between functions $g$ and $\widetilde{g}$ is

$$\delta^{g, \widetilde{g}} = \lambda_{\max} \left( \frac{1}{N} X^\top X - \frac{1}{\widetilde{N}} \widetilde{X}^\top \widetilde{X} \right).$$

Hence, we can take $\delta = \max(\delta^{g, g_1}, \ldots, \delta^{g, g_n})$.

Finally, we estimate the strong convexity parameter as $\mu = \lambda$.

As mentioned in the main body of the paper, we simulate the operation of 25 devices on one machine. For the synthetic dataset, samples on the workers are generated by adding unbiased Gaussian noise to the server data. For simulations with real data, we considered the LIBSVM datasets (a9a, w7a, w8a) and give each worker a full data. Then, each device selects at random a part of size $m$ from the full dataset. Some samples can occur on more than one worker (in this way we artificially increase the data size).

The parameters $L, \delta$ are estimated as written above. For the synthetic dataset we choose the noise level and the regularization parameter such that $L/\delta = 200$ and $L/\lambda = 10^5$. For the real datasets the regularization parameter is chosen such that $L/\lambda = 10^6$. In Table 2, we give all values of $L, \delta, m, \mu$.

Table 2: The value of the parameters $L, \delta, m, \mu$ in experiments.

| Dataset | $L$ | $\mu$ | $m$ | $\delta$ |
|---|---|---|---|---|
| synthetic | $10^4$ | $10^{-1}$ | — | 20 |
| a9a | $2 \cdot 10^5$ | $2 \cdot 10^{-1}$ | 5000 | 300 |
| w7a | $6,5 \cdot 10^4$ | $6,5 \cdot 10^{-2}$ | 7000 | 70 |
| w8a | $1,3 \cdot 10^5$ | $1,3 \cdot 10^{-1}$ | 10000 | 90 |