# OpenReview forum: "Optimal Gradient Sliding and its Application to Optimal Distributed Optimization Under Similarity"
_NeurIPS.cc/2022/Conference — NeurIPS 2022 Accept_

### Official Review · Reviewer_3MZq · 2022-07-06

**Rating:** 4
**Confidence:** 2
**Soundness:** 3 good
**Presentation:** 2 fair
**Contribution:** 2 fair

**Summary:**

This paper considers a gradient sliding approach for distributed optimization. The idea is that by interleaving the computations between past and future iterates, the distributed optimization system is more robust to noise and convergence can be accelerated.

**Questions:**

It is possible that all the rest of the issues are misunderstandings, and if we can clarify them (and discuss how they can be clarified in the paper), I can increase my score.

Broad strokes questions

 - Why is it called extragradient method? Extragradient usually implies taking two gradient calls and then using the second one, in order to get a "midpoint discretization" scheme and improve performance, but at a 2x overhead. None of the algorithms seem to be doing this.

 - It seems like the method mainly focuses on an outer loop / inner loop approach, where the outer loop achieves linear convergence and the inner loop borrows from other SOTA methods to get good guarantees. This seems appropriate for optimal rates, but I don't see the clear benefit to *distributed* optimization. Is this a noise variance controlling technique, or straggler mitigation technique? I guess what I'm asking is, if A = gradient sliding and B = optimal method, is this paper just A + B (arbitrary mix and match), or are there deeper ramifications for using this combination?

 - There is the big question also of what is hidden in the constants. For example, the scale of delta may be big in practice. More to the point, if delta is small in practice, to the point where the rates are tight, wouldn't the competitor methods work pretty well? A more extensive numerical experiments comparison would show if this is really a practical method.

 - What's the big idea behind using the prox operations in step 5 of alg 1? Is it just to achieve optimal rates? I ask because it seems prox operations are used, but aren't really discussed in terms of key benefits and numerical insights here, and the standard is usually just gradient descent.

Technical questions
 - In (8), I see a dependency of error on 1/T^2. However, in (9), T is a constant. This doesn't seem logical, since later inner iterations should require higher precision (and thus larger T) than initial ones. This of course affects the main final results (Thm 5)

 - Theorems 4 and 10 seem to better match the inner / outer loop issues, but they are kind of terrible rates then. How do they compare against competitors?


**Strengths And Weaknesses:**

Strengths
 - The paper does a pretty thorough related works review, and seems to be integrating sophisticated methods.

Weaknesses
 - It is a bit unwieldy to have 2 theorem boxes, which are referred to in the paper, in the appendix. In general I have a hard time understanding why each one is contributing something different, and something like this should be discussed more clearly in the main text.

 - The second set of numerical results are extremely inconclusive, and does not show consistent advantage at all.

---

> ### Author Response · Authors · 2022-08-01
> **Response to Reviewer 3MZq (part 3)**
>
>
> > **It seems like the method mainly focuses on an outer loop / inner loop approach**, where the outer loop achieves linear convergence and the inner loop borrows from other SOTA methods to get good guarantees. This seems appropriate for optimal rates, but I don't see the clear benefit to distributed optimization. Is this a noise variance controlling technique, or straggler mitigation technique? I guess what I'm asking is, if A = gradient sliding and B = optimal method, is this paper just A + B (arbitrary mix and match), or are there deeper ramifications for using this combination?
>
> 1) Note that all of the methods in Table 1 are “outer loop / inner loop approach”.
> To put it in Reviewer terms, the community has been solving $A + B$ since 2014, but has not gotten the optimal result during those 8 years. As we noted earlier, most of the papers in Table 1 were presented at A* conferences, and among the authors of these papers (often first and second authors) are very famous and highly cited scientists.
>
> 2) In answering the question (*Why is it called extragradient method?*) we tried to convey that the creation of $A$ required a great deal of effort.
>
> 3) *Is this a noise variance controlling technique, or straggler mitigation technique?* Neither, we explained the intuition of our method above. One can look for some connections our methods with variance controlling and straggler mitigation, but that would be artificial, we didn't put in these connections when we created the methods. Perhaps Reviewer sees something.
>
> > **There is the big question also of what is hidden in the constants** For example, the scale of delta may be big in practice. More to the point, if delta is small in practice, to the point where the rates are tight, wouldn't the competitor methods work pretty well? A more extensive numerical experiments comparison would show if this is really a practical method.
>
> 1) In the paragraph from line 43, we give an explanation why similarity is interesting to consider. This is primarily due to the fact that it is natural and has a good theoretical background. In particular, one can prove that if the data are uniformly distributed among the devices, then
> $$
> ||\nabla^2 f_i (x) - \nabla^2 f_j (x) || \sim \frac{L}{\sqrt{m}} ~~ \text{or}~~ \frac{L}{\sqrt{m}},
> $$
> where $L$ is a Lipshitz constant of gradients, $m$ is a size of local datasets on devices. What is more, this similarity is popular in literature.
>
> 2) *A more extensive numerical experiments comparison would show if this is really a practical method.* We have added more datasets for the first group of experiments. In the future we plan to add experiments for other practical problems, but for now due to the time limit can only provide this.
>
> > **What's the big idea behind using the prox operations in step 5 of alg 1?** Is it just to achieve optimal rates? I ask because it seems prox operations are used, but aren't really discussed in terms of key benefits and numerical insights here, and the standard is usually just gradient descent.
>
> 1) We probably have partially clarified the answer to the question when we discussed the other questions above.
>
> 2) It is important to note that the subproblem/prox operator in line 5 is solved by an additional optimizer. For example, here one can use gradient descent or accelerated gradient descent or other methods, the main thing is to find $x^{k+1}_f$ with necessary accuracy. But the accuracy of the solution is given by the unusual condition (4) (see Theorem 1), that is why we use a rather rare method (see Theorem 2), because it gives such guarantees that we need.
>
> > **In (8), I see a dependency of error on 1/T^2. However, in (9), T is a constant.** This doesn't seem logical, since later inner iterations should require higher precision (and thus larger T) than initial ones. This of course affects the main final results (Thm 5)
>
> $T$ is fixed, it also does not depend in any way on the iteration number, nor on the desired accuracy $\varepsilon$. The only thing we need to guarantee for solving the problem from line 5 of Algorithm 1 is that condition (4) from Theorem 1 is satisfied. And that condition is satisfied with fixed $T$ from (8).
>
> > **Theorems 4 and 10 seem to better match the inner / outer loop issues, but they are kind of terrible rates then.** How do they compare against competitors?
>
> Theorems 4,6, 10, 12 are given for the case when $\mu = 0$, this is the convex case for minimization and monotone for variational inequalities. And in these cases our algorithm  is also optimal and outperforms competitors in the application to the distributed problem with the similarity assumption.
>
> We don't quite understand what Reviewer meant by *Theorems 4 and 10 seem to better match the inner / outer loop issues*.  Please explain.

---

> > ### Comment · Reviewer_3MZq · 2022-08-03
> > **reply**
> >
> > Based on your responses, it seems that this paper may be trying to solve a seminal problem that I am less familiar with, and trying to achieve bounds, which your framework fits.
> >
> > To clarify where my questions/concerns come from, from a practical implementation point of view, I think it is still not clear to me that  the usual flavor of machine learning problems are well-solved using this approach (which it seems now has 3 loops if you account for the approximate prox calculation, and a pre-decided $T$, which from my experience these techniques tend to be pessimistic). Figure 1 is a nice inclusion, however.
> >
> > However, if the goal of the paper is to give a method that achieves a lower bound, which is a long-standing problem, that is a nice theoretical contribution. Since this is not my problem space, I would defer to the reviews of others more embedded in this problem area.

---

> > > ### Author Response · Authors · 2022-08-03
> > > **Response to Reviewer 3MZq**
> > >
> > > Thanks so much for the review and time!
> > >
> > > We greatly appreciate Reviewer's careful attention to our paper.
> > >
> > > If Reviewer feel that he/she are not an expert on the subject, we politely ask to let AC know about it and to take part in a discussion with other Reviewers. Please keep our paper in mind. Reviewer is the only one who rejects our paper and gives "fair" score of the contribution. We are very worried about our work, as probably Reviewer is about his/her submits.
> > >
> > > P.S. All our algorithms contain 2 loops: the first (main one) by iterator $k$, the second one by inexact calculation of the prox.

---

> ### Author Response · Authors · 2022-08-01
> **Response to Reviewer 3MZq (part 2)**
>
>
> > **Why is it called extragradient method?** Extragradient usually implies taking two gradient calls and then using the second one, in order to get a "midpoint discretization" scheme and improve performance, but at a 2x overhead. None of the algorithms seem to be doing this.
>
> To understand the intuitions of this method, let us try to lay out a complete picture of how it came about.
>
> 1) The problem of optimal distributed methods under similarity assumption has been known for a long time. It has been an open problem for 8 years. Works from Table 1 have been mostly published at leading conferences such as NeurIPS, ICML, AISTATS, ICLR. But these methods did not reach the lower bounds. These papers consider different basic ideas. We send Reviewer to Section 1.1 and 2 of [1] (Hadrien Hendrikx et al). There, one of the basic approaches to similarity methods is clearly outlined, as well as the history of the issue. It has become clear to us that we need to look at the problem of similarity methods differently. The idea of a composite problem seemed interesting. This is exactly what we describe in the beginning of the paper:
> $$
> r (x) = f_1(x) + \frac{1}{n} \sum_{i=1}^n [f_i(x) - f_1(x)]
> $$
> where $q(x) = f_1(x)$ is convex, and $p(x) = \frac{1}{n} \sum_{i=1}^n [f_i(x) - f_1(x)]$ may be non-convex.
>
> 2) The problem above is a composite problem. To obtain optimal estimates for such problems, sliding algorithms are usually used (see Section 1.2 of our work). At the highest level, the idea of sliding can be stated as follows (in theory it will be more difficult, but it gives an intuition), let us do a gradient descent as follows:
> $$
> x^{k+1} = x^k - \gamma (\nabla q (x^k) + \nabla p(w^k)),
> $$
> where we update the point $w^k$ quite rarely, for example, once every $t$ iterations $w^k = x^k$, otherwise it is just taken from the previous iteration $w^k = w^{k-1}$. This allows $\nabla p$ to be called very rarely.
> But for the problem "strongly convex = convex + possibly non-convex" no sliding was invented.
>
> 3) In parallel with this (without additional thoughts about similarity), we thought about sliding for composite variational inequalities (equation (10) in our paper):
> $$
> R(x) = Q(x) + P(x).
> $$
> To quickly understand the intuitions of variational inequalities, simply change to the operator on the gradient of the function: $R \to \nabla r$,  $Q \to \nabla q$, $P \to \nabla p$.
> The issue is that different methods are used for variational inequalities compared to minimization problems. Gradient descent for variational inequalities is not optimal. But the extragradient method is optimal:
> $$
> u^k = x^k - \gamma R(x^k), ~~~~ x^{k+1} = x^k - \gamma R(u^k).
> $$
> As a result, we got Algorithm 2 (see our paper). It combines the ideas of extragradient and sliding: in line 4, when we compute \tilde u^k we make sliding, because $P(x^k)$ is fixed, but $Q$ is changing. In line 5, we make an extra step with $R(u^k)$.
> Algorithm 2 has two interesting and important features: a) (12) – a rare stopping criterion for monotone variational inequalities, such a criterion is used only in one very recent paper (see Theorem 8 in our paper), b) in theoretical analysis it is not necessary to assume the monotonicity of one of the operators (monotonicity for minimization problems means convexity). The second feature is exactly what we needed for the method with similarity.
>
> 4) Then Algorithm 2 was accelerated using the Nesterov acceleration to produce Algorithm 1. Therefore, Algorithm 1 is called the accelerated extragradient.
>
> Here we have tried to outline very basic ideas and intuitions, it is easy to see that Algorithms 1 and 2 do not look as simple as we described them here. But hopefully we have given some insight into our thoughts. We would be happy to include some of these intuitions in the paper if we have extra space.
>
>
> [1] Hadrien Hendrikx, Lin Xiao, Sebastien Bubeck, Francis Bach, and Laurent Massoulie. Statisti- cally preconditioned accelerated gradient method for distributed optimization.

---

> > ### Comment · Reviewer_3MZq · 2022-08-03
> > **reply**
> >
> > This description was helpful. To avoid naming collisions within the community, I would not call it extragradient, since that is a different method, and just call it accelerated nesterov, or something like that.
> >
> > In point 3, does the VI framework also follow a strongly convex = convex + possibly nonconvex form? If so that would help unify the two stories; otherwise, they appear quite disparate.

---

> > > ### Author Response · Authors · 2022-08-03
> > > **Response to Reviewer 3MZq**
> > >
> > > > **To avoid naming collisions within the community, I would not call it extragradient, since that is a different method, and just call it accelerated nesterov, or something like that.**
> > >
> > > Accelerated Nesterov is the same method as Accelerated Gradient Descent. This is a method / acceleration technique invented by Yuri Nesterov in the second half of the 20th century. This does not fully reflect the idea of ​​our method. Recall that our method is "extragradient + sliding for a composite problem + Nesterov acceleration".
> > >
> > > In our algorithm's name, we wanted to emphasize that we accelerate the extragradient method. We thought about how we can highlight the fact that inside extragradient there is also sliding. But stopped on the current naming.
> > >
> > > We are absolutely open to changing the names of our methods.
> > >
> > > > **In point 3, does the VI framework also follow a strongly convex = convex + possibly nonconvex form? If so that would help unify the two stories; otherwise, they appear quite disparate.**
> > >
> > > Minimization problems are a special case of VIs (see line 200, Example 1). Therefore, yes, Algorithm 2 (for VIs) and point 3 of our response above is applicable for "strongly convex = convex + possibly nonconvex" minimization problems.
> > >
> > > Then, at first glance, it seems that we can consider the VIs and not consider the minimization problems. But it is not so. VIs are a broader class of problems than minimization problems, it is more complicated, in particular, it includes saddle point problems (min-max). Minimization problems can be solved faster than general VI problems. See e.g. Table 2 and compare our estimates and lower bounds for minimization and saddle problems:
> > >
> > > $$
> > > \text{For minimization: } \sqrt{\frac{\delta}{\mu}} \cdot \log 1/\varepsilon ~~~~ \text{ and } ~~~~ \sqrt{\frac{L}{\mu}} \cdot \log 1/\varepsilon.
> > > $$
> > >
> > > $$
> > > \text{For saddles: } \frac{\delta}{\mu} \cdot \log 1/\varepsilon ~~~~ \text{ and } ~~~~ \frac{L}{\mu} \cdot \log 1/\varepsilon.
> > > $$
> > >
> > > $$\sqrt{\frac{\delta}{\mu}} \leq \frac{\delta}{\mu} ~~~~ \text{ and } ~~~~  \sqrt{\frac{L}{\mu}} \leq \frac{L}{\mu}$$.
> > >
> > > Estimates for minimization problems are better, so we consider them separately.

---

> ### Author Response · Authors · 2022-08-01
> **Response to Reviewer 3MZq (part 1)**
>
> We thank Reviewer **3MZq**  for review, time and the insightful comments, which will help to improve our work.
>
> > **It is a bit unwieldy to have 2 theorem boxes**, which are referred to in the paper, in the appendix. In general I have a hard time understanding why each one is contributing something different, and something like this should be discussed more clearly in the main text.
>
> Sorry, we don’t quite understand what Reviewer meant in that comment. Please explain, in more detail, it seems that the problem is not too big and we can calmly solve/explain it.
>
> > **The second set of numerical results are extremely inconclusive**, and does not show consistent advantage at all.
>
> We moved them to Appendix (see the revision). Instead, we added more datasets for the first group of experiments in the main part.

---

> > ### Comment · Reviewer_3MZq · 2022-08-03
> > **reply**
> >
> > In general these two comments are about what goes in the main paper and what goes in the appendix. My general rule of thumb is that I do not need the appendix to decide whether a paper warrants acceptance, and only use it to check for correctness, or if there are extra curiosities that are interesting but not essential. If an algorithm is claimed to be a major contribution, it needs to be in the main paper. If the claim is that numerical results are not a key contribution, and have moved them to the appendix, that is acceptable.

---

> > > ### Author Response · Authors · 2022-08-03
> > > **Response to Reviewer 3MZq**
> > >
> > > Thanks for reply, hope we understand now!
> > >
> > > In the main paper, we focus on strongly-convex/strongly-monotone problems when the parameter $\mu > 0$ (Assumptions 1 and 7). In Table 2 we present a comparison of the results in this case only. Algorithm 1 and Algorithm 2 are also algorithms for strongly-convex/strongly-monotone problems.
> > >
> > > Algorithm 3 and 4 (Appendix) are algorithms for the convex problem ($\mu = 0$). They differ slightly from Algorithms 1 and 2 because, for example, Nesterov acceleration is done differently for the strongly convex and convex cases. In the main article, we present the results in the convex case for completeness (to emphasize that we also have them along with the strongly-convex/strongly-monotone case).

---

### Official Review · Reviewer_21CM · 2022-07-10

**Rating:** 6
**Confidence:** 3
**Soundness:** 3 good
**Presentation:** 2 fair
**Contribution:** 3 good

**Summary:**

The paper combines extra gradient, gradient sliding, and acceleration to propose an accelerated extragradient sliding methods for convex composite optimization that achieves improved gradient complexity. When applied to distributed optimization problem under agents' function similarity, the method achieves lower bounds on communication and gradient complexities. The method is further extended to variational inequality setting.

**Questions:**

1. Line 23, "a network of $m$ agents." For Equation (2), is it $1/m$ or $1/n$? See also Line 47, line 45 line 56.

2. Line 44, why one would use the second order information for similarity notation? What would happen if one uses first/zeroth-order information for similarity notation. How to verify that function similarity holds.

3. If there is similarity, does it reduce back to the stochastic setting? For example, if using first-order similarity notation, one may rewrite the problem as a stochastic optimization with a discrete distribution and bounded variance of the first-order information.

4. For second-order similarity, how to evaluate it numerically?

5. The proposed method combines acceleration, extragradient, and gradient sliding. It is unclear which component improves the complexity bound, comparing to the existing literature. Please specify with a more detailed comparison with existing literature on the methodology.

## Minor Comments:
1. Line 6, 7, missing bracket.
2. Line 23, is it $m$ or $n$?
3. Line 29, there are $m$ agents or $m$ samples?
4. Line 30, what does it mean by mismatch between parameter and sample?
5. Assumption 1 with $\mu=0$ is normally treated as convexity while $\mu<0$ is weakly convex in the literature. It would be nice to stay consistent.
6. Example 2 Line 201, is it $\min_z$ or $\max_z$ as it is a saddle point problem.
7. Line 114,  120, missing bracket

**Ethics Review Area:**

["I don’t know"]

**Limitations:**

N.A.

**Strengths And Weaknesses:**

## Strength

1. The paper combines extra gradient, gradient sliding, and acceleration that achieves $O(\sqrt{L_p/\mu})$ and $O(\sqrt{L_q/\mu})$ gradient call for $\nabla p$ and $\nabla q$ respectively, improving over the condition number for minimization problem. It well suits the unbalanced problem.
2. The method is further extends to obtain optimal gradient sliding for VIs.
3. When applied to distribution optimization and distributed minimax optimization under similarity assumption, the proposed methods achieve the lower bounds on both communication and gradient complexities.


## Weakness

1. Presentation needs serious improvements. See questions below.

2. The numerical results are relatively less informative. On real datasets, the proposed method only achieves comparable performance with the state-of-the-art method.

---

> ### Author Response · Authors · 2022-08-01
> **Response to Reviewer 21CM (part 3)**
>
> > **Line 6, 7, missing bracket.**
>
> Fixed, thanks!
>
> > **Line 23, is it $m$ or $n$?**
>
> $n$. fixed!
>
> > **Line 29, there are m  agents or m  samples?**
>
> $m$ samples
>
> > **Line 30, what does it mean by mismatch between parameter and sample?**
>
> Thanks! We changed it to avoid misundertandings. Now there is “ the loss of the model x on the sample z^j_i”
>
> > **Assumption 1 with $\mu=0$  is normally treated as convexity while $\mu<0$  is weakly convex in the literature. It would be nice to stay consistent.**
>
> We deleted “weakly”. Thanks!
>
> > **Example 2 Line 201, is it $\min_z$ or $max_z$ as it is a saddle point problem.**
>
> $\max_z$. Fixed, thanks!
>
> > **Line 114, 120, missing bracket**
>
> Fixed, thanks!

---

> ### Author Response · Authors · 2022-08-01
> **Response to Reviewer 21CM (part 2)**
>
>
> > **The proposed method combines acceleration, extragradient, and gradient sliding. It is unclear which component improves the complexity bound, comparing to the existing literature. Please specify with a more detailed comparison with existing literature on the methodology.**
>
> If we are talking about our results for the distributed problem under similarity conditions, then the idea is based on 4 main things:
>
> 1)  consider the distributed problem as a composite problem, where one function is convex and the other is possibly nonconvex,
>
> 2) special sliding for composite variational inequalities based on the extragradient
>
> 3) the feature of this sliding is that it is applicable to the composite problem with monotone+nonmonotone operators (convex+nonconvex functions)
>
> 4) acceleration of this sliding.
>
> Next, in details. To understand the whole intuitions of our methods, let us try to lay out a complete picture of how it came about.
>
> 1) The problem of optimal distributed methods under similarity assumption (as we stated above) has been known for a long time. It has been an open problem for 8 years. Works from Table 1 have been mostly published at leading conferences such as NeurIPS, ICML, AISTATS, ICLR. But these methods did not reach the lower bounds. These papers consider different basic ideas. We send Reviewer to Section 1.1 and 2 of [1] (Hadrien Hendrikx et al). There, one of the basic approaches to similarity methods is clearly outlined, as well as the history of the issue. It has become clear to us that we need to look at the problem of similarity methods differently. The idea of a composite problem seemed interesting. This is exactly what we describe in the beginning of the paper:
> $$
> r (x) = f_1(x) + \frac{1}{n} \sum_{i=1}^n [f_i(x) - f_1(x)]
> $$
> where $q(x) = f_1(x)$ is convex, and $p(x) = \frac{1}{n} \sum_{i=1}^n [f_i(x) - f_1(x)]$ may be non-convex.
>
> 2) The problem above is a composite problem. To obtain optimal estimates for such problems, sliding algorithms are usually used (see Section 1.2 of our work). At the highest level, the idea of sliding can be stated as follows (in theory it will be more difficult, but it gives an intuition), let us do a gradient descent as follows:
> $$
> x^{k+1} = x^k - \gamma (\nabla q (x^k) + \nabla p(w^k)),
> $$
> where we update the point $w^k$ quite rarely, for example, once every $t$ iterations $w^k = x^k$, otherwise it is just taken from the previous iteration $w^k = w^{k-1}$. This allows $\nabla p$ to be called very rarely.
> But for the problem "strongly convex = convex + possibly non-convex" no sliding was invented.
>
> 3) In parallel with this (without additional thoughts about similarity), we thought about sliding for composite variational inequalities (equation (10) in our paper):
> $$
> R(x) = Q(x) + P(x).
> $$
> To quickly understand the intuitions of variational inequalities, simply change to the operator on the gradient of the function: $R \to \nabla r$,  $Q \to \nabla q$, $P \to \nabla p$.
> The issue is that different methods are used for variational inequalities compared to minimization problems. Gradient descent for variational inequalities is not optimal. But the extragradient method is optimal:
> $$
> u^k = x^k - \gamma R(x^k), ~~~~ x^{k+1} = x^k - \gamma R(u^k).
> $$
> As a result, we got Algorithm 2 (see our paper). It combines the ideas of extragradient and sliding: in line 4, when we compute \tilde u^k we make sliding, because $P(x^k)$ is fixed, but $Q$ is changing. In line 5, we make an extra step with $R(u^k)$.
> Algorithm 2 has two interesting and important features: a) (12) – a rare stopping criterion for monotone variational inequalities, such a criterion is used only in one very recent paper (see Theorem 8 in our paper), b) in theoretical analysis it is not necessary to assume the monotonicity of one of the operators (monotonicity for minimization problems means convexity). The second feature is exactly what we needed for the method with similarity.
>
> 4) Then Algorithm 2 was accelerated using the Nesterov acceleration to produce Algorithm 1. Therefore, Algorithm 1 is called the accelerated extragradient.
>
> Here we have tried to outline very basic ideas and intuitions, it is easy to see that Algorithms 1 and 2 do not look as simple as we described them here. But hopefully we have given some insight into our thoughts. We would be happy to include some of these intuitions in the paper if we have extra space.
>
>
> [1] Hadrien Hendrikx, Lin Xiao, Sebastien Bubeck, Francis Bach, and Laurent Massoulie. Statisti- cally preconditioned accelerated gradient method for distributed optimization.

---

> ### Author Response · Authors · 2022-08-01
> **Response to Reviewer 21CM (part 1)**
>
> We thank Reviewer **21CM**  for review, time and the insightful comments, which will help to improve our work.
>
> Next, we answer the questions asked by Reviewer.
>
> > **Presentation needs serious improvements. See questions below.**
>
> We improved. Please see the revision and answers below.
>
> > **The numerical results are relatively less informative. On real datasets, the proposed method only achieves comparable performance with the state-of-the-art method.**
>
> We added more real datasets in the revision. Our method wins competitors on these datasets.
>
> > **Line 23, "a network of $m$ agents." For Equation (2), is it $1/m$ or $1/n$ ? See also Line 47, line 45 line 56.**
>
> The correct option in line 23: “$n$ agents”. We fixed, thanks! Lines 45, 47, 56 are correct
>
> > **Line 44, why one would use the second order information for similarity notation?** What would happen if one uses first/zeroth-order information for similarity notation. How to verify that function similarity holds.
>
> 1) After line 44, we give an explanation why the second-order similarity is interesting to consider. This is primarily due to the fact that it is natural and has a good theoretical background. In particular, one can prove that if the data are uniformly distributed among the devices, then
> $$
> ||\nabla^2 f_i (x) - \nabla^2 f_j (x) || \sim \frac{L}{\sqrt{m}} ~~ \text{or}~~ \frac{L}{\sqrt{m}},
> $$
> where $L$ is a Lipshitz constant of gradients, $m$ is a size of local datasets on devices.
> the second-order similarity is popular in literature. All of the papers in Table 1 address it; most of them presented at NeuIPS, ICML, ICLR, and AISTATS. But the most interesting thing is that for 8 years, none of the papers could reach the lower bounds. This was a challenge for us.
>
> 2) If we consider first-order similarity, this, too, has its place in the literature, but it is a different setting of the problem with a different background and other work-competitors. Moreover, we do not know the same facts about first-order similarity that also naturally emerged as for second-order similarity, i.e. we mean theorems in the spirit of "if data are uniformly distributed over the devices, then the constant of first-order similarity is...". The only thing we can prove something from the second order similarity:
> if $||\nabla^2 f_i (x) - \nabla^2 f_j (x) || \leq \delta$ for all x, then the function $(f_i - f_j)$ has a $\delta$-Lipshitz gradient:
> $$
> || \nabla f_i (x)  - \nabla f_j (x) - \nabla f_i (y) + \nabla f_j (y)|| \leq \delta || x - y||
> $$
> From this equation we have that
> $$
> || \nabla f_i (x)  - \nabla f_j (x)|| - || \nabla f_i (y) - \nabla f_j (y) || \leq || \nabla f_i (x)  - \nabla f_j (x) - \nabla f_i (y) + \nabla f_j (y)||  \leq \delta || x - y||$$
> and
> $$
> || \nabla f_i (x)  - \nabla f_j (x)|| \leq \delta || x - y|| + || \nabla f_i (y) - \nabla f_j (y) ||
> $$
> Let us fix some $y = y_0$ and assume that $|| \nabla f_i (y_0) - \nabla f_j (y_0) || \leq G$, it means that
> $$
> || \nabla f_i (x)  - \nabla f_j (x)|| \leq \delta || x - y|| + G
> $$
> If we solve the problem on $R^d$, it means that the first-order similarity parameter can be very large (because $|| x - y||$ is unbounded).
>
> > **If there is similarity, does it reduce back to the stochastic setting?** For example, if using first-order similarity notation, one may rewrite the problem as a stochastic optimization with a discrete distribution and bounded variance of the first-order information.
>
> It seems possible to consider that we work with stochastic hessians (using exactly the same reasoning that Reviewer gave for gradients). This raises a number of questions
>
> 1) Most methods for similarity, including ours, use only gradients. Methods that use hessian are quite expensive, especially in a distributed setup, when hessians need to be sent to the server. Moreover, Newton's methods are good for local convergences, while we solve a global minimization problem.
>
> 2) We also ask Reviewer to pay attention to Table 1, most methods of competitors using hessians (column “Order”) give bad rates (and some even do not converge at all).
>
>
> > **For second-order similarity, how to evaluate it numerically?**
>
> 1)  If we can guarantee that the data is uniformly distributed among the computing devices, then the estimate we discussed above is valid $\delta \sim L/\sqrt{m}$. For example, if one solves a problem on a computing cluster and distributes the data independently and uniformly.
>
> 2) If we do not know how the data are distributed on the devices, then we can only measure this similarity numerically. The easiest way is to send the hessian to the server a couple of times, but this is expensive. So we can use practical tricks. For example, send only the diagonal elements of a hessian, or compress it (send the top-10% of the largest hessian values).

---

> ### Comment · Reviewer_21CM · 2022-08-09
> **Post rebuttal comment**
>
> Thanks for the detailed reply. I do not have any more questions. I would like to raise my score.

---

> > ### Author Response · Authors · 2022-08-09
> > **Thank you!**
> >
> > We greatly thank Reviewer **21CM** for the response, important comments, and positive final feedback!

---

### Official Review · Reviewer_disq · 2022-07-11

**Rating:** 7
**Confidence:** 4
**Soundness:** 3 good
**Presentation:** 2 fair
**Contribution:** 3 good

**Summary:**

Paper considers convex minimization of sum of smooth functions r(x) = p(x) + q(x) with Lipschtiz-smoothness constants Lp and Lq (Lp < Lq). Here they assume that p can be non-convex and q is convex. This paper provides a gradient sliding algorithm which only uses O(\sqrt{Lp}) gradients of p and O(\sqrt{Lq}) gradients of q. This kind of separation is useful when first order oracle of p is much more costlier than that of q. Paper provides these results when r is convex, or strongly convex.

This algorithm is then used to obtain optimal communication and gradient complexities for distribution finite sum optimization under similarity Hessian assumption. Paper also extends the sliding scheme to the setting of solving a monotonic variation inequality of sum of two Lipschitz continuous operators.

**Questions:**

1. Why does the algorithm work? Why is this called “accelerated extragradient”
2. What are examples where r is mu-strongly convex, p is nonconvex and q is convex?  It looks like the one in simulations doesn’t satisfy this. If there are no such examples, can step 5 of algorithm be solved using accelerated smooth strongly convex optimization?
3. Function similarity assumption and the assumption 6 different (notice j in former). What is the relation between the two and why does the paper use a different assumption?
4. Acc SONATA [44], what is the complexity to solve proximal steps? Is it O(\sqrt{L_i/\mu} \log(1/eps)) because of strong convexity? If yes, this should be noted in the table as well.

5. In experiments, what are the r, p and q, and Lp, Lq and \mu?
6. In experimental section what does lambda=0,1 mean? What was used for the experiments and why were these choices made?
7. Please provide comparison to AccSONATA which has the SOTA communication complexity. The sub problems should be easy to compute if they are convex or strongly-convex.
8. In Synthetic minimization dataset how much noise was added?

Minor comments

9. Lq >= Lp >= mu assumption should be more prominant
10. “Weakly convex” function also has another meaning of mu < 0. May be use “convex”?


**Limitations:**

No limitations are provided. Authors may discuss the challenges of extending results to to nonconvex settings

**Strengths And Weaknesses:**

Strengths
- Gradient sliding results seems novel
- Results seem to pass quick checking of logic
- Paper provides results for two different settings and applies them to distributed optimization to obtain optimal guarantees
- Experiments section

Weakness
- Paper does not explain the intuition behind the algorithm. The exposition should also make it clear how they are related to known ideas. These help the community assimilate the results and build on top of them to produce future scientific output.
- No proper examples are provided for the settings studied in the paper.
- Not enough experiment details are provided and there seems to be missing baselines
- It is not clear whether the missing entries in the “Local gradient complexity” column of the table actually means these oracles are hard to solve. The best complexity for solving the oracle sub-problems should be mentioned here.

After rebuttal
- Authors addressed almost all of my concerns. Assuming authors will take action on my suggestions, including the new ones about the intuition and rewording the claim, I am increasing my score.

---

> ### Author Response · Authors · 2022-08-01
> **Response to Reviewer disq (part 3)**
>
> ​​
> > **Why does the algorithm work? Why is this called “accelerated extragradient”**
>
> We answered this question above (*Paper does not explain the intuition behind the algorithm.*)
>
> > **What are examples where r is mu-strongly convex, p is nonconvex and q is convex? It looks like the one in simulations doesn’t satisfy this. If there are no such examples, can step 5 of algorithm be solved using accelerated smooth strongly convex optimization?**
>
> We answered this question above (*No proper examples are provided for the settings studied in the paper.*)
>
> > **Function similarity assumption and the assumption 6 different (notice j in former). What is the relation between the two and why does the paper use a different assumption? **
>
> Thanks! We fixed Assumption 6 in the revision. It is easy to prove that if $||\nabla^2 f_j (x)  - \nabla^2 f_i (x)|| \leq \delta$ for all $i,j$ and $x$, then
> $$
> || \nabla^2 f (x) - \nabla^2 f_j (x)|| = || \frac{1}{n} \sum _{i=1}^n [\nabla^2 f_i (x) - \nabla^2 f_j (x) ] || \leq \frac{1}{n} \sum _{i=1}^n || \nabla^2 f_i (x) - \nabla^2 f_j (x)|| \leq \delta.
> $$
>
> > **Acc SONATA [44], what is the complexity to solve proximal steps? Is it O(\sqrt{L_i/\mu} \log(1/eps)) because of strong convexity? If yes, this should be noted in the table as well.**
>
> We answered this question above (*It is not clear whether the missing entries in the “Local gradient complexity”*).
>
> > **In experiments, what are the r, p and q, and Lp, Lq and \mu?**
>
> In experiments we use Algorithm 1 with $r,p,q$ as described in (3) and in Section 3. We added details about $L,\mu,\delta$ (see the revision).
>
> > **In experimental section what does lambda=0,1 mean? What was used for the experiments and why were these choices made?**
>
> Thanks! That is an inaccuracy. The regularization $\lambda = 0.1$ from (17) is correct only for synthetic dataset. We corrected this in the revision. The full selection of the constant is described as follows:
>
> "In Section C.2  we explain how the parameters $L$ and $\delta$ are estimated.
> For the synthetic dataset we choose the noise level and the regularization parameter such that $L/\delta = 200$ and $L/\lambda = 10^5$.
> For the real datasets the regularization parameter is chosen such that $L/\lambda = 10^6$."
>
> > **Please provide comparison to AccSONATA which has the SOTA communication complexity. The sub problems should be easy to compute if they are convex or strongly-convex.**
>
> We added AccSONATA (see revision). But it does not give good results. This is most likely due to the fact that it uses envelope/Catalyst acceleration, which in practice works worse than Nesterov's direct acceleration.
>
> > **In Synthetic minimization dataset how much noise was added?**
>
> As we noted earlier, the full selection of the constant is described as follows:
>
> "In Section C.2  we explain how the parameters $L$ and $\delta$ are estimated.
> For the synthetic dataset we choose the noise level and the regularization parameter such that $L/\delta = 200$ and $L/\lambda = 10^5$.
> For the real datasets the regularization parameter is chosen such that $L/\lambda = 10^6$."
>
>
> > **Lq >= Lp >= mu assumption should be more prominant**
>
> Sorry, we don't quite understand what was meant. In fact, we can get the same results as in Theorem 3 for the case when $L_q < L_p$ - see Line 161. But from the point of view of the similarity application we are interested only in the case when $L_p \leq L_q$, i.e. $\delta \leq L$.
>
> > **“Weakly convex” function also has another meaning of mu < 0. May be use “convex”?**
>
> Thanks, we fixed it - see the revision.

---

> > ### Comment · Reviewer_disq · 2022-08-07
> > **Reply (part 3)**
> >
> > Thanks for the details! I am satisfied with most of the answers. I think experimental section is still confusing. For example, in the the updated paper, I am still not clear what values were used for lambda of synthetic experiments. I suggest providing a table of the problem and algorithm parameters used for different settings.

---

> > > ### Author Response · Authors · 2022-08-08
> > > **Response to Reviewer disq. Reply (part 3)**
> > >
> > > For the synthetic dataset $L \approx 10^4$, $\lambda \approx 0.1$, $\delta \approx 50$.
> > >
> > > We added Table 2 with all values of $L$, $\mu = \lambda$ and $\sigma$ - see Appendix C.2 in the revision.
> > >
> > > In line 271 -273 we described that we use parameters from theory:
> > >
> > > "The settings of the methods are made as described in the original papers. For algorithms that assume an absolutely accurate solution of local problems (DANE, SPAG, AccSONATA), we use AcGD with an accuracy of $10^{-12}$ as a subsolver."

---

> ### Author Response · Authors · 2022-08-01
> **Response to Reviewer disq (part 2)**
>
>
> > **No proper examples are provided for the settings studied in the paper.**
>
> If we understand correctly, this sentence is explained through the following question (if we are wrong, please clarify).
>
> > **What are examples where r is mu-strongly convex, p is nonconvex and q is convex? It looks like the one in simulations doesn’t satisfy this. If there are no such examples, can step 5 of algorithm be solved using accelerated smooth strongly convex optimization?**
>
> 1) First of all, the problem for which our sliding was invented suits these assumptions. This problem is the distributed minimization problem under similarity. See (3) in our paper:
> $$
> r (x) = f_1(x) + \frac{1}{n} \sum_{i=1}^n [f_i(x) - f_1(x)],
> $$
> where $q(x) = f_1(x)$ is convex, and $p(x) = \frac{1}{n} \sum_{i=1}^n [f_i(x) - f_1(x)]$ may be non-convex.
> ​​More specifically, for example, our experimental setup: $r(x)$ is a distributed regression problem with a regularizer (strongly convex), $q(x) = f_1(x)$ is a server loss function (it is also strongly convex, but it is enough that it is convex for theory), $p(x) = \frac{1}{n} \sum_{i=1}^n [f_i(x) - f_1(x)]$ (non-convex).
>
> 2) In fact, sliding can also be applied to “convex + convex” problems (such problems satisfy the "convex + not necessarily convex" assumption). Such problems are quite common. One interesting and popular example at the moment is the so-called personalized federated learning [2] and [3]
>
> [2] Filip Hanzely, Peter Richtárik, Federated Learning of a Mixture of Global and Local Models
>
> [3] Filip Hanzely, Slavomír Hanzely, Samuel Horváth, Peter Richtárik. Lower Bounds and Optimal Algorithms for Personalized Federated Learning
>
> 3) *Can step 5 of algorithm be solved using accelerated smooth strongly convex optimization?*
> The answer is yes anyway, because the problem in line 5 is convex anyway (convexity depends on the function $q$, and it is convex). But we use not the accelerated gradient method, because we need to guarantee condition (4) (see Theorem 1) when solving the internal problem, it is important for obtaining optimality. For the accelerated gradient method, we did not find such convergence results.
>
> >**Not enough experiment details are provided and there seems to be missing baselines**
>
> We expanded the experimental section in the revision. Please indicate which details Reviewer would like to see more of.
>
>
> **It is not clear whether the missing entries in the “Local gradient complexity” column of the table actually means these oracles are hard to solve. The best complexity for solving the oracle sub-problems should be mentioned here.**
>
> Dashes in Table 1 means that the authors assume they know how to solve any local subproblems with accuracy $\varepsilon = 0$. That is why we did not include such results in Table 1. We don’t know practical algorithms that can solve minimization problems absolutely precisely. Reviewer suggests including estimates for accelerated gradient descent in Table 1, but this is not really fair to other competitors; we do not know how the authors' method will behave if we add inaccuracy to their method, especially this issue concerns 2nd-order methods. Some of the papers have puzzled over this issue and indicated what method, with what accuracy and with what parameters we need  to use to solve the local subproblems.
>
> The question of stopping the internal method for local subproblems is also important for us because we use a rather unusual stopping criterion and this has turned out to be one of the key places of our theoretical analysis and an important stepping stone to achieving optimality in local computations.

---

> > ### Comment · Reviewer_disq · 2022-08-07
> > **Reply (part 2)**
> >
> > I am satisfied with the most of the above responses. Please summarize them in the manuscript to improve the presentation.
> >
> > It would great if the authors can point out which paper does the following
> > > Some of the papers have puzzled over this issue and indicated what method, with what accuracy and with what parameters we need to use to solve the local subproblems.
> >
> > > Reviewer suggests including estimates for accelerated gradient descent in Table 1, but this is not really fair to other competitors; we do not know how the authors' method will behave if we add inaccuracy to their method, especially this issue concerns 2nd-order method
> >
> > Acc SONATA is not a 2nd-order method. Separately, I feel there might a trivial analysis of it to allow for approximate proximal point operators. I don't understand what authors mean by "this is not really fair to other competitors". Scientific progress is not supposed to be a game :). Essence of my question was whether there is a simple analysis of approximate Acc SONATA. If yes, what would be the rate. If not, why is this analysis hard.

---

> > > ### Author Response · Authors · 2022-08-08
> > > **Response to Reviewer disq. Reply (part 2)**
> > >
> > >
> > > > **Acc SONATA is not a 2nd-order method. Separately, I feel there might a trivial analysis of it to allow for approximate proximal point operators. I don't understand what authors mean by "this is not really fair to other competitors". Scientific progress is not supposed to be a game :). Essence of my question was whether there is a simple analysis of approximate Acc SONATA. If yes, what would be the rate. If not, why is this analysis hard.**
> > >
> > > We filled all empty cells in Table 1 (see the revision). As Reviewer asked, we assume that we solve the proximal subproblem inexactly using Accelerated gradient method with precision $\varepsilon$ (this is reflected in the footnote to Table 1).
> > >
> > > > **I am satisfied with the most of the above responses. Please summarize them in the manuscript to improve the presentation.**
> > >
> > > We are glad to hear that, the basic edits Reviewer asked for we made during the rebuttal. We are ready to add the rest when we have extra space, e.g to add and to discuss composite problems from [2,3].
> > >
> > > 2] Filip Hanzely, Peter Richtárik, Federated Learning of a Mixture of Global and Local Models
> > >
> > > [3] Filip Hanzely, Slavomír Hanzely, Samuel Horváth, Peter Richtárik. Lower Bounds and Optimal Algorithms for Personalized Federated Learning

---

> ### Author Response · Authors · 2022-08-01
> **Response to Reviewer disq (part 1)**
>
> We thank Reviewer **disq**  for review, time and the insightful comments, which will help to improve our work.
>
> Next, we answer the questions asked by Reviewer.
>
> **Paper does not explain the intuition behind the algorithm. The exposition should also make it clear how they are related to known ideas. These help the community assimilate the results and build on top of them to produce future scientific output.**
>
> To understand the intuitions of this method, let us try to lay out a complete picture of how it came about.
>
> 1) The problem of optimal distributed methods under similarity assumption has been known for a long time. It has been an open problem for 8 years. Works from Table 1 have been mostly published at leading conferences such as NeurIPS, ICML, AISTATS, ICLR. But these methods did not reach the lower bounds. These papers consider different basic ideas. We send Reviewer to Section 1.1 and 2 of [1] (Hadrien Hendrikx et al). There, one of the basic approaches to similarity methods is clearly outlined, as well as the history of the issue. It has become clear to us that we need to look at the problem of similarity methods differently. The idea of a composite problem seemed interesting. This is exactly what we describe in the beginning of the paper:
> $$
> r (x) = f_1(x) + \frac{1}{n} \sum_{i=1}^n [f_i(x) - f_1(x)]
> $$
> where $q(x) = f_1(x)$ is convex, and $p(x) = \frac{1}{n} \sum_{i=1}^n [f_i(x) - f_1(x)]$ may be non-convex.
>
> 2) The problem above is a composite problem. To obtain optimal estimates for such problems, sliding algorithms are usually used (see Section 1.2 of our work). At the highest level, the idea of sliding can be stated as follows (in theory it will be more difficult, but it gives an intuition), let us do a gradient descent as follows:
> $$
> x^{k+1} = x^k - \gamma (\nabla q (x^k) + \nabla p(w^k)),
> $$
> where we update the point $w^k$ quite rarely, for example, once every $t$ iterations $w^k = x^k$, otherwise it is just taken from the previous iteration $w^k = w^{k-1}$. This allows $\nabla p$ to be called very rarely.
> But for the problem "strongly convex = convex + possibly non-convex" no sliding was invented.
>
> 3) In parallel with this (without additional thoughts about similarity), we thought about sliding for composite variational inequalities (equation (10) in our paper):
> $$
> R(x) = Q(x) + P(x).
> $$
> To quickly understand the intuitions of variational inequalities, simply change to the operator on the gradient of the function: $R \to \nabla r$,  $Q \to \nabla q$, $P \to \nabla p$.
> The issue is that different methods are used for variational inequalities compared to minimization problems. Gradient descent for variational inequalities is not optimal. But the extragradient method is optimal:
> $$
> u^k = x^k - \gamma R(x^k), ~~~~ x^{k+1} = x^k - \gamma R(u^k).
> $$
> As a result, we got Algorithm 2 (see our paper). It combines the ideas of extragradient and sliding: in line 4, when we compute \tilde u^k we make sliding, because $P(x^k)$ is fixed, but $Q$ is changing. In line 5, we make an extra step with $R(u^k)$.
> Algorithm 2 has two interesting and important features: a) (12) – a rare stopping criterion for monotone variational inequalities, such a criterion is used only in one very recent paper (see Theorem 8 in our paper), b) in theoretical analysis it is not necessary to assume the monotonicity of one of the operators (monotonicity for minimization problems means convexity). The second feature is exactly what we needed for the method with similarity.
>
> 4) Then Algorithm 2 was accelerated using the Nesterov acceleration to produce Algorithm 1. Therefore, Algorithm 1 is called the accelerated extragradient.
>
> Here we have tried to outline very basic ideas and intuitions, it is easy to see that Algorithms 1 and 2 do not look as simple as we described them here. But hopefully we have given some insight into our thoughts. We would be happy to include some of these intuitions in the paper if we have extra space.
>
>
> [1] Hadrien Hendrikx, Lin Xiao, Sebastien Bubeck, Francis Bach, and Laurent Massoulie. Statisti- cally preconditioned accelerated gradient method for distributed optimization.

---

> > ### Comment · Reviewer_disq · 2022-08-07
> > **Reply (part 1)**
> >
> > I really appreciate that the authors provided the motivation for their problem and some details about the origin the algorithm.
> >
> > However, I still don't understand the step 5 and 6 of Algorithm 1. Specifically, I am hoping to understand how q(x) is used as an approximately proximable function step 5, but its gradient is directly used in algorithms 1. And why this works out.
> >
> > Let me ask a counter question to understand the algorithm better. Could you have obtained the same results using the following approximate version of the accelerated proximal gradient descent method [Algorithm 1, APGM]?
> >
> > 4. $y_{k} = (1 - \tau_k) x_k + \tau_k z_k$
> >
> > 5. $z_{k+1} \approx \arg\min_z p(y_k) + \langle \nabla p(y_k), z - y_k\rangle + \frac{\gamma_k}2 \|\|z- y_k\|\|^2 + q(x)$ (similar to step 5 of current algorithm)
> >
> > 6. $x_{k+1} = (1 - \tau_k) x_k + \tau_k z_{k+1}$
> >
> > I feel the above algorithm is more common and easier to understand than step 5 and 6 of this paper's algorithm. Can the authors explain what is the advantage of their algorithm over the above approximate APGM?
> >
> > [APGM] Tseng, Paul. "On accelerated proximal gradient methods for convex-concave optimization." submitted to SIAM Journal on Optimization 2.3 (2008). https://www.mit.edu/~dimitrib/PTseng/papers/apgm.pdf

---

> > > ### Author Response · Authors · 2022-08-08
> > > **Response to Reviewer disq. Reply (part 1)**
> > >
> > > Thanks for the response!
> > >
> > > 1) Line 5 from the response is typical for similarity algorithms. Reviewer can open the papers from Table 1, all papers solve such or similar proximal subproblems. For convenience, we give the basic idea (idea of DANE, 2014, see Table 1) that most works use (DANE, DANE-HB, DANE-LS, AIDE, SONATA, SPAG, AccSONATA). This idea use the proximal gradient descent with the Bregman divergence (Reviewer can read it in Section 1.1 of [1]):
> > > $$
> > > x^{k+1} = \arg \min_x \left( \langle \nabla r (x^k), x - x^k\rangle + \frac{1}{\eta} D_{\varphi} (x, x^k) \right),
> > > $$
> > > where $D_{\varphi} (x, x^k)$ is the Bregman divergence:
> > > $$
> > > D_{\varphi} (x, x^k) = \varphi(x) - \varphi(x^k) - \langle \nabla \varphi(x^k), x - x^k \rangle.
> > > $$
> > > The key is to use
> > > $$
> > > \varphi(x) = f_1(x) + \frac{\delta}{2} || x ||^2,~~~ ~~~ \eta = 1.
> > > $$
> > > Then, we have the following iteration of the proximal gradient descent
> > > $$
> > > x^{k+1} = \arg \min_x \left( f_1(x) + \langle \nabla (r - f_1) (x^k), x \rangle + f_1(x) + \frac{\delta}{2} || x - x^k||^2 \right)
> > > $$
> > > If we rewrite it in terms of $q = f_1$ and $p = f - f_1$, we get
> > > $$
> > > x^{k+1} = \arg \min_x \left( q(x) + \langle \nabla p (x^k), x \rangle  + \frac{\delta}{2} || x - x^k||^2 \right)
> > > $$
> > > That's exactly what Reviewer wrote in line 5 of the response.
> > > For the method we just described, one can only prove the following estimates:
> > > $$
> > > O\left( \frac{\delta}{\mu} \log 1/e\right),
> > > $$
> > > which is not optimal.
> > >
> > > There is a reasonable question, but whether it is possible to speed up this approach. For example, there are accelerated versions of the proximal method (APGM or others).
> > >
> > > But the literature since 2014 has been unable to add direct acceleration to the proximal gradient method for the similarity problem and to obtain optimal rates. This is due to the fact that the analysis of accelerated methods requires convexity of both the function $p$ and the function $q$. This is not the case for the similarity problem (see below in the rebuttal that our problem is "non-convex+concave=strongly convex"). Analysis for the non-accelerated version (we described) were invented back around 2014, but the acceleration has become a challenge for the community. It seems to us that any team taking on the similarity task has tried to accelerate the idea described above (we have also tried).
> > >
> > > There were attempts to accelerate with a heavy ball (DANE-HB), but they only give the optimal rate for quadratic problems. There were attempts to accelerate with Catalyst envelope (AIDE, SPAG, AccSONATA), but they work only for some cases or don't meet lower bounds. Moreover, direct acceleration is a more practical trick; acceleration with Catalyst-type envelops often performs worse in practice and gives suboptimal rates.
> > >
> > > 2) We can finalize the above as follows:
> > >
> > > the proximal gradient descent + direct acceleration (Nesterov's acceleration or Tseng's variant) = no results, problems in analysis (need convexity of both functions $p$ and $q$)
> > >
> > > the proximal gradient descent + heavy ball = optimal only for quadratic problems
> > >
> > > the proximal gradient descentd + Catalyst = no optimal rates
> > >
> > > 3) Our work is not based on gradient descent. It is based on the extragradient, adds sliding (proximal calculations) to it, and then accelerate it. For more details, see our rebuttal above.
> > >
> > > We hope we were able to explain!
> > >
> > > P.S. Reviewer is right that we use inexact proximal calculations, but we call it "sliding" (and not only us - see Section 1.2 of our paper), often other varieties of sliding that can be found for different types of problems do similar things (inexact proximal calculations). This technique is most likely called "sliding" because of the physicality of the name, because we "slide" only one function while the gradient of the other function is fixed.
> > >
> > >
> > >
> > > [1] Hadrien Hendrikx, Lin Xiao, Sebastien Bubeck, Francis Bach, and Laurent Massoulie. Statistically preconditioned accelerated gradient method for distributed optimization.

---

### Official Review · Reviewer_xSX2 · 2022-07-12

**Rating:** 7
**Confidence:** 4
**Soundness:** 3 good
**Presentation:** 3 good
**Contribution:** 3 good

**Summary:**

The author studied a convex optimization problem, where the objective function can be decomposed into a smooth convex function and a possible nonconvex function. There are three major contributions: 1. the author proposed gradient sliding algorithm which achieves optimal complexity of gradient calls of each component functions; 2. the author proved the algorithm achieves optimal communication complexity under $\delta$-similarity; 3. the author applied the proposed method to a class of distributed saddle-point optimization problem.

**Questions:**

I am wondering what is the author's thoughts on the comparison between the proposed methods and the so-called local SGD studied in [1],
and any thoughts on the proposed methods for deep learning.

[1] Stich, Sebastian U. "Local SGD converges fast and communicates little." arXiv preprint arXiv:1805.09767 (2018).

**Ethics Review Area:**

["I don’t know"]

**Limitations:**

the authors have adequately addressed the limitations

**Strengths And Weaknesses:**

The paper is a good one with strong theoretical results and clear presentation.

---

> ### Author Response · Authors · 2022-08-01
> **Response to Reviewer xSX2**
>
> We thank Reviewer **xSX2**  for review, time and the insightful comments, which will help to improve our work.
>
> Next, we answer the questions asked by Reviewer.
>
> > **I am wondering what is the author's thoughts on the comparison between the proposed methods and the so-called local SGD studied in [1]**
>
> Let us highlight some of the differences:
>
> 1) In works about local methods (in particular [1]) all devices work in parallel, in our method only server works, other devices rarely send required gradients.
>
> 2) In local methods, for example in Local SGD, when local steps happen, in fact local function on device f_i is minimized. In our case the local subproblem looks different (see line 5 in Algorithm 1 of our paper).
>
> 3) Our method is primarily sharpened to use similarity (although it can be used for any problem), local methods were presented for general problems. Therefore our method works better (optimal) for problems under hessian similarity.
>
> [1] Stich, Sebastian U. "Local SGD converges fast and communicates little." arXiv preprint arXiv:1805.09767 (2018).
>
> > **any thoughts on the proposed methods for deep learning**
>
> This is a very interesting area of research. We are busy with it now and do not want to reveal all the details, but we tell some ideas:
>
> In theoretical works it is often assumed that neural networks have a $L$-Lipschitz gradient. One can also recall that the learning rate/step in theoretical analysis is usually $\gamma \sim 1/L$. Combining these two facts and knowing some classical values of learning rate $\gamma_{classical}$ for training e.g. ResNet on ImageNet, one can find out the estimate of the Lipschitz constant of gradients for the ResNet training problem on ImageNet: $L_{est} \sim 1/\gamma_{classical}$. If we train our model in a distributed manner and divide the data uniformly, there is a similarity $\delta$ between workers. Again from theory this $\delta$ can be estimated as $\delta_{est} \sim L_{est}/\sqrt{m}$, where $m$ is a local data size. It turns out that some modifications of our algorithms (from the current paper under review) work for distributed training of neural networks (if we use $\delta_{est} \sim L_{est}/\sqrt{m}$ as a parameter).

---

> > ### Comment · Reviewer_xSX2 · 2022-08-09
> > **Major contribution of clarification**
> >
> > Thank you for the explanation. Here I am wondering:
> >
> > 1. whether gradient sliding techniques have been applied previously in distributed optimization?
> >
> > 2. If yes to question 1, then could you kindly give more intuition why those can not achieve optimal results while yours can as in the current work you claimed to solve a long standing problem. Thanks.

---

> > > ### Author Response · Authors · 2022-08-09
> > > **Response to Reviewer xSX2 (Major contribution of clarification)**
> > >
> > > Thank you for the response!
> > >
> > > At the moment we are discussing this with Reviewer **disq**.
> > > Please, read "Response to Reviewer disq (part 1)", "Reply (part 1)" by Reviewer **disq**, "Response to Reviewer disq. Reply (part 1)".
> > >
> > > For convenience, we duplicate it here.
> > >
> > > 1) Reviewer **disq** noted that we use inexact proximal calculations (see line 5 go Algorithm 1), but we call it "sliding" (and not only us - see Section 1.2 of our paper), often other varieties of sliding that can be found for different types of problems do similar things (inexact proximal calculations). This technique is most likely called "sliding" because of the physicality of the name, because we "slide" only one function while the gradient of the other function is fixed.
> > >
> > > 2) Sliding/inexact proximal calculations is used in all the papers on similarity. Reviewer can open the papers from Table 1, all papers solve proximal subproblems. For convenience, we give the basic idea (idea of DANE, 2014, see Table 1) that most works use (DANE, DANE-HB, DANE-LS, AIDE, SONATA, SPAG, AccSONATA). This idea use the proximal gradient descent with the Bregman divergence (Reviewer can read it in Section 1.1 of [1]):
> > > $$
> > > x^{k+1} = \arg \min_x \left( \langle \nabla r (x^k), x - x^k\rangle + \frac{1}{\eta} D_{\varphi} (x, x^k) \right),
> > > $$
> > > where $D_{\varphi} (x, x^k)$ is the Bregman divergence:
> > > $$
> > > D_{\varphi} (x, x^k) = \varphi(x) - \varphi(x^k) - \langle \nabla \varphi(x^k), x - x^k \rangle.
> > > $$
> > > The key is to use
> > > $$
> > > \varphi(x) = f_1(x) + \frac{\delta}{2} || x ||^2,~~~ ~~~ \eta = 1.
> > > $$
> > > Then, we have the following iteration of the proximal gradient descent
> > > $$
> > > x^{k+1} = \arg \min_x \left( f_1(x) + \langle \nabla (r - f_1) (x^k), x \rangle + f_1(x) + \frac{\delta}{2} || x - x^k||^2 \right)
> > > $$
> > > If we rewrite it in terms of $q = f_1$ and $p = f - f_1$, we get
> > > $$
> > > x^{k+1} = \arg \min_x \left( q(x) + \langle \nabla p (x^k), x \rangle  + \frac{\delta}{2} || x - x^k||^2 \right)
> > > $$
> > > For the method we just described, one can only prove the following estimates:
> > > $$
> > > O\left( \frac{\delta}{\mu} \log 1/e\right),
> > > $$
> > > which is not optimal.
> > >
> > > There is a reasonable question, but whether it is possible to speed up this approach. For example, there are accelerated versions of the proximal method.
> > >
> > > But the literature since 2014 has been unable to add direct acceleration to the proximal gradient method for the similarity problem and to obtain optimal rates. This is due to the fact that the analysis of accelerated methods requires convexity of both the function $p$ and the function $q$. This is not the case for the similarity problem (see (3), our problem is "non-convex+concave=strongly convex"). Analysis for the non-accelerated version (we described) were invented back around 2014, but the acceleration has become a challenge for the community. It seems to us that any team taking on the similarity task has tried to accelerate the idea described above (we have also tried).
> > >
> > > There were attempts to accelerate with a heavy ball (DANE-HB), but they only give the optimal rate for quadratic problems. There were attempts to accelerate with Catalyst envelope (AIDE, SPAG, AccSONATA), but they work only for some cases or don't meet lower bounds. Moreover, direct acceleration is a more practical trick; acceleration with Catalyst-type envelops often performs worse in practice and gives suboptimal rates.
> > >
> > > 3) We can finalize the above as follows:
> > >
> > > the proximal gradient descent + direct acceleration= no results, problems in analysis (need convexity of both functions $p$ and $q$)
> > >
> > > the proximal gradient descent + heavy ball = optimal only for quadratic problems
> > >
> > > the proximal gradient descentd + Catalyst = no optimal rates
> > >
> > > 3) Our work is not based on gradient descent. It is based on the extragradient, adds sliding (proximal calculations) to it, and then accelerate it. Please read "Response to Reviewer disq (part 1)" for more details about the whole idea of our method.
> > >
> > > We hope we were able to explain!
> > >
> > > [1] Hadrien Hendrikx, Lin Xiao, Sebastien Bubeck, Francis Bach, and Laurent Massoulie. Statistically preconditioned accelerated gradient method for distributed optimization.

---

### Author Response · Authors · 2022-08-01
**Rebuttal Revision**

Dear Reviewers, Area Chairs and Senior Area Chairs!

We published a revision of our paper in which we have tried to solve most of the issues. Сhanges are highlighted in blue. What is new:

1) We deleted the word “weakly”, now if $\mu = 0$, then the function is “convex” (without “weakly”). Thanks Reviewer **disq**, **21CM**!

2) We changed Assumptions 6 and 12 a bit to have the same definition of similarity in the introduction and in the main part. Thanks Reviewer **disq**! It does not affect the results.

3) We fixed typos. Thanks Reviewer **21CM**!

4) We made changes to the experimental section:

a) added two more datasets in the first group of experiments (asked by Reviewer **3MZq**),

b) added AccSONATA (asked by Reviewer **disq**),

c) added more details about constant $L,\mu, \delta$ choice and estimates (asked by Reviewer **disq**),

d) moved the second set of experiments to Appendix C.1, because of space limit.

Thank you very much for your work! You really helped make our paper better.

---

### Author Response · Authors · 2022-08-07
**A kind reminder about rebuttals**

With this message, we would just like to kindly remind Reviewers that we would be happy if Reviewers would participate in the rebuttal discussion process. We are looking forward to hearing from Reviewers **xSX2**, **disq** and **21CM**. We thank Reviewer **3MZq** for the response to the rebuttal.

---

### Meta-Review · Area_Chair_8nzu · 2022-08-26

**Recommendation:** Accept
**Confidence:** Less certain

**Metareview:**

The paper extends gradient sliding to the situation where both functions are smooth and the sum is strongly convex. The resulting algorithm is then applied to distributed optimization settings and similarity assumptions, where it jointly achieves optimal gradient evaluations and communication complexities, improving on prior complexity bounds by logarithmic factors.

Initially, the reviewers were unclear about the motivation and construction of the algorithm, as well as the significance of the theoretical results. However, through extensive discussion, most of the issues were clarified to the satisfaction of the reviewers. Consequently, I recommend acceptance of the paper and urge the reviewers to carefully incorporate all the clarifications in their rebuttal into the camera ready paper.

In addition, please provide an accurate answer (either yes or no) to question 3a in the reproducibility checklist.

**Award:**

No

---

### Decision · Program_Chairs · 2022-09-14

Accept